# Behavioral Entropy-Guided Dataset Generation for Offline Reinforcement Learning

**Wesley A. Suttle**\*, **Aamodh Suresh**\*, **Carlos Nieto-Granda**

U.S. Army Research Laboratory
Adelphi, MD 20783, USA
`wesley.a.suttle.ctr@army.mil`,
`aamodh@gmail.com`,
`carlos.p.nieto2.civ@army.mil`

## ABSTRACT

Entropy-based objectives are widely used to perform state space exploration in reinforcement learning (RL) and dataset generation for offline RL. Behavioral entropy (BE), a rigorous generalization of classical entropies that incorporates cognitive and perceptual biases of agents, was recently proposed for discrete settings and shown to be a promising metric for robotic exploration problems. In this work, we propose using BE as a principled exploration objective for systematically generating datasets that provide diverse state space coverage in complex, continuous, potentially high-dimensional domains. To achieve this, we extend the notion of BE to continuous settings, derive tractable $k$-nearest neighbor estimators, provide theoretical guarantees for these estimators, and develop practical reward functions that can be used with standard RL methods to learn BE-maximizing policies. Using standard MuJoCo environments, we experimentally compare the performance of offline RL algorithms for a variety of downstream tasks on datasets generated using BE, Rényi, and Shannon entropy-maximizing policies, as well as the SMM and RND algorithms. We find that offline RL algorithms trained on datasets collected using BE outperform those trained on datasets collected using Shannon entropy, SMM, and RND on all tasks considered, and on 80% of the tasks compared to datasets collected using Rényi entropy.

## 1 INTRODUCTION

Reinforcement learning (RL) methods can successfully solve challenging tasks in complex environments, even outperforming humans in a variety of cases (Mnih et al., 2015; Silver et al., 2018). However, due to the online nature of standard RL algorithms and their reliance on informative, often hand-engineered reward signals, RL methods are typically sample-inefficient and lack generalizability to new tasks. Offline RL (Levine et al., 2020; Prudencio et al., 2023) is an alternative approach that applies RL-based techniques to train policies entirely offline using static datasets of trajectories collected from the target domain. The key innovation is that a single offline dataset can be relabeled with a variety of different reward functions, enabling reuse of datasets to learn a variety of downstream tasks. This paradigm magnifies the importance of learning to generate datasets with diverse coverage of the state space in hopes of covering regions that correspond to a wide a variety of downstream tasks Yarats et al. (2022). The design of existing algorithms for dataset generation (Pathak et al., 2017; Eysenbach et al., 2018; Lee et al., 2019; Burda et al., 2019; Liu & Abbeel, 2021; Yarats et al., 2021) relies on uncertainty metrics, such as entropy, to quantify the quality and control the variety of state space coverage. This renders the choice of uncertainty metrics critical to the diversity of datasets that can be achieved. While Shannon entropy (SE) and Rényi entropy (RE) have been widely used as exploration objectives in RL Hazan et al. (2019); Liu & Abbeel (2021); Yarats et al. (2021); Zhang et al. (2021); Yuan et al. (2022), the variety of datasets that can be achieved using them is limited. In the SE case this is due to the fact that SE provides just a single objective and thus a single notion of optimal coverage (see Figure 2b). In the RE case, though its parametric form provides a variety of notions of coverage, this coverage remains partial (see Figure 2b) and comes at

---

\*Equal contribution.

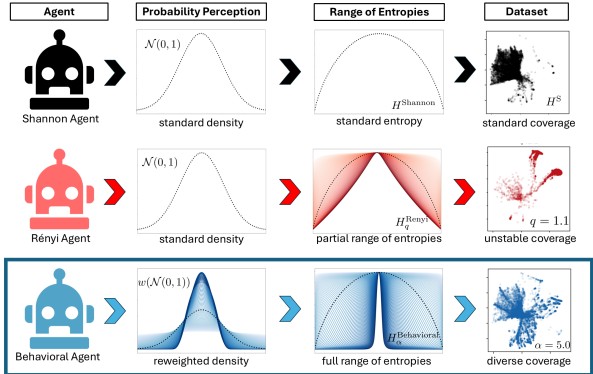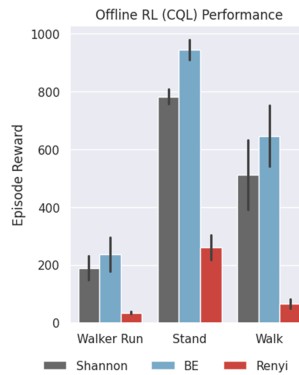

Figure 1: **(Left)** Comparison of Shannon entropy, Rényi entropy, and behavioral entropy (ours) and their effects on dataset generation, shown in PHATE plots, when used as an exploration objective. **(Right)** Performance comparison of an offline RL algorithm (CQL) for three downstream tasks on datasets generated using Shannon, behavioral entropy (ours), and Rényi entropy for the parameter $q = 1.1$ shown in the left-hand figure.

the cost of instability arising due to its inherent discontinuity in parameter space (Suresh et al., 2024) and poor coverage for many parameter values (see Figure 1). Developing a family of exploration objectives overcoming these issues remains an open question.

Recently, Suresh et al. (2024) proposed *behavioral entropy* (BE), a novel generalization of classical entropies that composes SE with the probability weighting functions widely used in behavioral economics to model human decision making (Dhami, 2016). In Suresh et al. (2024), the authors rigorously established two key facts: (i) BE is a valid generalization of the classical notion of entropy, and (ii) BE provides the most general notion of entropy to date in the sense that its parametric form captures a wider range of valid generalized entropies than existing parametric families of entropies, such as RE. The first property guarantees BE is smooth in its parameter and attains its maximum on the uniform distribution, which is critical for exploration and data generation applications aiming for uniform coverage. The second property stems from BE's definition in terms of probability weighting functions (Prelec, 1998), the use of which allows it to achieve a broader range of entropies than comparable methods (see Figure 1). As Suresh et al. (2024) experimentally demonstrated on robotic exploration problems, when BE is used as an objective this flexibility induces diverse exploration policies ranging from those providing coarse and widespread coverage of the environment to dense and focused coverage, achieving the best performance and the widest variety of coverages in discrete probability spaces compared with SE and RE. Together, these results establish BE as a principled alternative to existing exploration objectives that provides a rich variety of exploration behaviors and state space coverage. However, the current formulation of BE is restricted to discrete probability distributions and the experimental evaluation provided in Suresh et al. (2024) is limited to classical, planning-based solutions to discretized, two-dimensional robotic exploration tasks. This impedes the applicability of BE to more complex problems.

In this paper we propose using BE as a principled exploration objective for systematically generating datasets that provide diverse state space coverage in complex, continuous, potentially high-dimensional domains. We hypothesize that using BE in this way will lead to superior offline RL performance on BE-generated datasets compared with objectives providing an inferior diversity of coverage, particularly SE and RE, but also the widely used Random Network Distillation (RND) (Burda et al., 2019) and State Marginal Matching (SMM) (Lee et al., 2019) algorithms. There are three primary challenges to using BE in this way: (i) a continuous-spaces version of BE must be formulated; (ii) principled estimators for BE that enjoy theoretical guarantees while remaining tractable in continuous, potentially high-dimensional settings must be developed; (iii) practical RL methods for training BE-maximizing policies must be derived. We address all three of these challenges in this work, then use the resulting RL method to generate datasets providing a rich variety of state space coverage for subsequent offline RL. We experimentally confirm our hypothesis, demonstrating that BE-generated datasets lead to superior offline RL performance over SE, RE, RND, and SMM datasets, and that offline RL methods enjoy better data- and sample-efficiency when applied to BE- and RE-generated datasets compared with existing benchmarks.

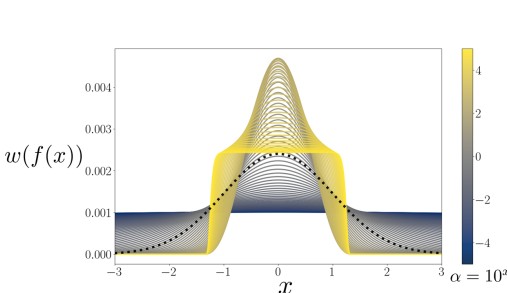
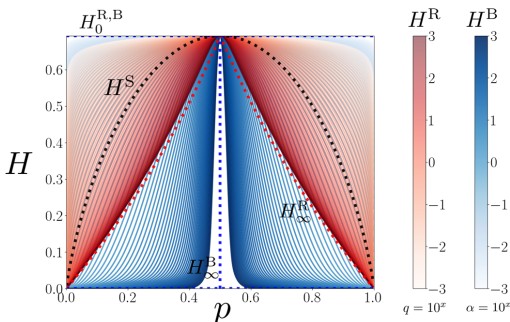

(a) **Perception of normal distribution ($\mathcal{N}(0,1)$) under Prelec weighting function.** Black dotted line shows standard normal density, $f$. For $\alpha \approx 0$ the perceived density is uniform, indicating *over-weighting* of uncertainty over entire support of $f$. For $\alpha \gg 0$ the perceived density approaches a step function around the mean, indicating *under-weighting* of uncertainty near $f$'s tails. $\alpha = 1$ recovers original $f$.

(b) **Diversity of Shannon, Rényi and BE values as a function of Bernoulli trial parameter $p$.** Behavioral entropy $H^B$ captures entire behavior spectrum from overvaluing uncertainty (light blue, $\alpha \approx 0$) to highly undervaluing uncertainty (dark blue, $\alpha \gg 0$). Rényi, $H^R$, captures the former (light red, $q \approx 0$), but cannot capture the latter. Dotted red curve shows $H^R$ as $q \to \infty$. Shannon, $H^S$, is dotted black curve.

Figure 2: Visualizations of probability weightings (left) and superior expressiveness of BE (right).

Our main contributions are:

- **Behavioral entropy estimation in continuous spaces.** We propose a version of BE applicable to continuous probability distributions, derive $k$-nearest neighbor ($k$-NN) estimators for BE with general probability weighting functions, and provide convergence guarantees and probabilistic bounds characterizing the bias and variance of these estimators.
- **Exploration and data generation via RL-based BE maximization.** We derive practical BE-maximizing exploration objectives and experimentally illustrate their effectiveness to generate datasets with diverse levels of state space coverage in unsupervised RL settings.
- **Offline RL performance on BE-generated data.** We experimentally evaluate the performance of offline RL algorithms for a variety of downstream tasks on BE, RE, SE, RND, and SMM datasets. We find that BE datasets lead to superior offline RL performance over SE, RE, RND, and SMM, and that offline RL methods enjoy better data- and sample-efficiency when applied to BE- and RE-generated datasets compared with existing benchmarks.

## 2 BEHAVIORAL ENTROPY IN CONTINUOUS SPACES

**Behavioral Entropy.** The various notions of entropy that have been studied in the literature since the initial work by Shannon (1948) quantify the uncertainty inherent in a random variable by measuring how evenly distributed its associated probability density is over its support. Let $X$ be a discrete random variable over a finite set of $M$ elements, and let $p$ denote its probability mass function (p.m.f.). Two classical and widely used entropies are the Shannon and Rényi entropies (Shannon, 1948; Rényi, 1961), given respectively by

$$H^S(X) = -\sum_{i=1}^{M} \log(p_i)p_i, \qquad (1) \qquad H_q^R(X) = \frac{1}{1-q}\log\sum_{i=1}^{M} p_i^q, \; q > 0, q \neq 1. \quad (2)$$

These entropy functionals, along with others such as Tsallis entropy (Tsallis, 1988), belong to the class of *admissible generalized entropies* satisfying the first three Shannon-Khinchin axioms (see (Amigó et al., 2018) for details). These axioms ensure that an entropy functional is well-behaved by ensuring their continuity, stability with respect to addition or removal of known outcomes, and maximality of the uniform distribution.

As mentioned in the introduction, the recent work (Suresh et al., 2024) proposed composing Shannon's entropy with the probability weighting functions widely used in behavioral economics to encode human cognitive and perceptual biases. The composition of Shannon's entropy with Prelec's probability weighting function Prelec (1998) yielded BE, an admissible generalized entropy that

provides a tractable mathematical formalism for incorporating helpful human biases into uncertainty quantification via entropy. The probability weighting functions used to encode human biases are defined as follows (see (Dhami, 2016, Ch. 2.2)).

**Definition 1.** *A function $w : [0,1] \to [0,1]$ is said to be a **probability weighting function** if it is continuous, strictly increasing, satisfies $w(0) = 0$ and $w(1) = 1$, and has a unique, continuous, and strictly increasing inverse.*

The most widely used probability weighting function, and that which was the focus of Suresh et al. (2024), is Prelec's probability weighting, given by

$$w(x) = e^{-\beta(-\log x)^\alpha}, \quad \alpha, \beta > 0. \tag{3}$$

Prelec's function is smooth in both the probability and parameter space and has the ability to control the fixed point and shape of the weighting function (see (Prelec, 1998; Dhami, 2016) for details). Figure 2a illustrates its effect on a Gaussian density. Equipped with equation 3, Suresh et al. (2024) proposed and studied the following.

**Definition 2.** *Letting $w$ be as in equation 3, **behavioral entropy** is given by*

$$H^{\text{B}}(X) = -\sum_{i=1}^{M} w(p_i) \log(w(p_i)). \tag{4}$$

The parameter $\alpha$ controls the shape of the perceived probability curve, enabling a wide range of probability perceptions (see Figure 2a) and the resulting perceived entropies, as illustrated in Figure 2b. Under the condition that $\beta = e^{(1-\alpha)\log(\log(M))}$, equation 4 was shown in Suresh et al. (2024) to belong to the class of admissible generalized entropies. With this conditioning, equation 3 allows control of the third fixed point of $w$, which is critical for ensuring BE remains an admissible entropy, whereas other probability weighting functions lack this property (Dhami, 2016) and do not generate meaningful, admissible generalized entropies.

Interestingly, it was shown in (Suresh et al., 2024) that using BE over other approaches led to significant acceleration of robotic exploration tasks as well as emergent search behaviors similar to breadth-first and depth-first search, depending on choice of $\alpha$. These results indicate that BE holds promise as an objective for a broad range of exploration tasks in complex environments, yet Suresh et al. (2024) only applied BE to discrete, binary random variables. To pave the way for the application of BE to more complex problems, we now extend it to continuous settings.

**Differential Behavioral Entropy.** In this subsection we extend the definition of BE to that of differential BE, providing a novel entropy functional that is applicable in continuous, potentially high-dimensional spaces. Let $f \in \Delta(\mathcal{X})$ be a p.d.f. over $\mathcal{X} \subset \mathbb{R}^d$, where $d \in \mathbb{N}^+$. We first recall the differential versions of Shannon's entropy and Rényi entropy of order $q$, where $q > 0, q \neq 1$, which are defined, respectively, by

$$H^S(f) = -\int_{\mathcal{X}} \log(f(x))f(x)dx, \quad (5) \qquad H_q^R(f) = \frac{1}{1-q} \log \int_{\mathcal{X}} f^q(x)dx. \quad (6)$$

We will use these expressions in our experimental comparisons below and thus include them here for easy reference. We next state the definitions of our continuous-spaces analogues of equation 4.

**Definition 3.** *For an arbitrary probability weighting function $w$, **differential generalized behavioral entropy** is given by*

$$H^{B,w}(f) = -\int_{\mathcal{X}} \log(w(f(x)))w(f(x))dx. \tag{7}$$

*In particular, substituting Prelec's probability weighting from equation 3 into equation 7 yields **differential behavioral entropy**, given by*

$$H^{B,\alpha,\beta}(f) = \beta \int_{\mathcal{X}} e^{-\beta(-\log(f(x)))^\alpha}(-\log f(x))^\alpha dx. \tag{8}$$

It is important to note that, unlike Definition 1 for probability weightings in the discrete setting, where $w : [0,1] \to [0,1]$ in the continuous setting $w$ must be generalized to $w : [0,\infty) \to [0,\infty)$ to accommodate arbitrary densities, $f$. Desirable structural properties of $w$ are described in the detailed statement of Theorem 2 in the appendix. We will henceforth abuse both terminology and notation by omitting "differential" when referring to differential entropies and by suppressing the dependence of equation 8 on $\alpha, \beta$ when these are clear from context.

## 3 $k$-NEAREST NEIGHBOR BEHAVIORAL ENTROPY ESTIMATION

We next turn to the problem of BE estimation in continuous, potentially high-dimensional spaces. To accomplish this, we derive $k$-nearest neighbor ($k$-NN) estimates of BE along the lines of the estimates studied in (Kozachenko & Leonenko, 1987; Singh et al., 2003; Leonenko et al., 2008; Sricharan et al., 2012; Singh & Póczos, 2016) and others for Shannon and Rényi entropy. The $k$-NN family of nonparametric estimators enables estimation of arbitrary densities from a finite number of i.i.d. samples in continuous and potentially high-dimensional settings, making them particularly well-suited to the RL context considered in the following section. Let $f \in \Delta(\mathbb{R}^d)$ be a probability density function (p.d.f.) over $\mathbb{R}^d$, where $d \in \mathbb{R}^+$. Let $X_1, X_2, \ldots, X_n \sim f(\cdot)$ be $n \in \mathbb{N}^+$ i.i.d. samples drawn from $f$. In (Loftsgaarden & Quesenberry, 1965; Devroye & Wagner, 1977) and subsequent works it was established that, for suitably chosen $n$ and $k$, a reasonable approximation of $f$ is provided by the $k$-NN density estimator

$$\hat{f}(x) = \frac{k\Gamma(d/2+1)}{n\pi^{d/2}R_{k,n}^d(x)}, \tag{9}$$

where $R_{k,n}(x) = \|x - NN_k(x)\|_2$ is the Euclidean distance between $x$ and its $k$th nearest neighbor among $\{X_1, \ldots, X_n\}$ and $\Gamma(x) = \int_0^\infty t^{x-1}e^{-t}dt$ is the gamma function. When $x = X_i$, we will write $R_{i,k,n} = R_{k,n}(X_i)$ for simplicity. A natural first approximation to Shannon entropy of $f$ is given by the plug-in estimator

$$\widehat{H}_{k,n}^S(f) = -\frac{1}{n}\sum_{i=1}^n \log \hat{f}(X_i) \approx \mathbb{E}_{X_i \sim f(\cdot)}\left[-\log \hat{f}(X_i)\right] = -\int_{\mathbb{R}^n} \log \hat{f}(x) \cdot f(x)dx. \tag{10}$$

With this in mind, the naïve approach to estimating equation 7 for general $w$ is via

$$\widetilde{H}_{k,n}^{B,w}(f) = -\frac{1}{n}\sum_{i=1}^n w(\hat{f}(X_i))\log w(\hat{f}(X_i)). \tag{11}$$

Since $X_i \sim f(\cdot)$, however, the estimator of equation 11 is biased, since

$$\widetilde{H}_{k,n}^{B,w}(f) \approx \mathbb{E}_{X_i \sim f(\cdot)}\left[-w(\hat{f}(X_i))\log w(\hat{f}(X_i))\right] = -\int_{\mathbb{R}^n} w(\hat{f}(x))\log w(\hat{f}(x)) \cdot f(x)dx. \tag{12}$$

Dividing by the approximation $\hat{f}$ yields the alternative, importance sampling-corrected estimator

$$\widehat{H}_{k,n}^{B,w}(f) = -\frac{1}{n}\sum_{i=1}^n \frac{1}{\hat{f}(X_i)}w(\hat{f}(X_i))\log w(\hat{f}(X_i)) \tag{13}$$

$$\approx \mathbb{E}_{X_i \sim f(\cdot)}\left[\frac{1}{\hat{f}(X_i)}w(\hat{f}(X_i))\log w(\hat{f}(X_i))\right] = -\int_{\mathbb{R}^n} w(\hat{f}(x))\log w(\hat{f}(x))\frac{f(x)}{\hat{f}(x)}dx.$$

Though this importance sampling-corrected plug-in estimator will be biased for the same reasons detailed in (Singh et al., 2003, Thm. 8) for $\widehat{H}_{k,n}^S(f)$, for large $n$ and suitable $k$ equation 13 provides a reasonable estimator of equation 8, as characterized by the following results.

**Theorem 1.** *Suppose that $k := k_n \to \infty$, $\frac{k_n}{n} \to 0$, and $\frac{k_n}{\log n} \to \infty$ as $n \to \infty$. Assume that $w$ is Lipschitz, that $f$ is absolutely continuous, and that there exist $c_1, c_2 > 0$ such that $0 < c_1 \leq f(x) \leq c_2 < \infty$, for all $x \in \mathcal{X}$. Then $\widehat{H}_{k,n}^{B,w}(f) \to H^{B,w}(f)$ both uniformly and in probability.*

The result follows from the strong uniform consistency of $\hat{f}$ (Devroye & Wagner, 1977).

Though asymptotic guarantees like Theorem 1 are somewhat reassuring, in practice we will use finite $k$ in our $k$-NN estimators and therefore need a more fine-grained characterization of the bias and variance. Unfortunately, for finite $k$, the approximator $\widehat{H}_{k,n}^{B,w}(f)$ remains biased as $n \to \infty$ due to the biasedness of $\hat{f}$ for fixed $k$ and the lack of a known bias correction procedure for our BE approximator $\widehat{H}_{k,n}^{B,w}(f)$. This contrasts with the situation for simpler estimators like those for Shannon and Rényi entropies, for which explicit bias correction terms are known (see Singh et al. (2003); Leonenko et al. (2008); Singh & Póczos (2016)). Nonetheless, we establish probabilistic guarantees on the bias and variance of our proposed BE estimator in the following main result.

**Theorem 2.** *Suppose $f$ and $w$ satisfy suitable differentiability and continuity conditions. Then, for $\varepsilon > 0$, it holds with probability $1 - \varepsilon$ that*

$$\left| \mathbb{E}\left[ \widehat{H}_{k,n}^{B,w}(f) \right] - H^{B,w}(f) \right| = \begin{cases} \mathcal{O}\left( \frac{k}{n} \right)^{\frac{\xi}{d}} + \mathcal{O}\left( \left( \frac{k}{n} \right)^{\frac{2}{d}} \log n + \sqrt{\frac{\log(n/\varepsilon)}{k}} \right) & \text{if } d > 2, \\ \mathcal{O}\left( \frac{k}{n} \right)^{\frac{\xi}{d}} + \mathcal{O}\left( \frac{k}{n} \log n + \sqrt{\frac{\log(n/\varepsilon)}{k}} \right) & \text{if } d = 1, 2, \end{cases} \tag{14}$$

$$Var\left( \widehat{H}_{k,n}^{B,w}(f) \right) = \mathcal{O}\left( \frac{1}{n} \right). \tag{15}$$

The proof and a precise statement are provided in the appendix. Equipped with the $k$-NN estimators and theoretical guarantees derived above, we next turn to their application in the RL setting.

## 4 BEHAVIORAL ENTROPY IN THE REINFORCEMENT LEARNING CONTEXT

In this section we leverage the $k$-nearest neighbor generalized behavioral entropy estimates developed in the previous section for general probability weightings to derive a practical reward function that can be used in conjunction with standard RL methods to maximize behavioral entropy of an RL agent's state occupancy measure.

**Behavioral Entropy as an RL Objective.** State occupancy measure entropy has been used as an exploration objective for RL in a wide range of previous works (Hazan et al., 2019; Liu & Abbeel, 2021; Yarats et al., 2021; Zhang et al., 2021; Yuan et al., 2022). In order to formally define behavioral entropy for state occupancy measures in this context, we first provide preliminary background on Markov decision processes (MDPs). Let an average-reward MDP $\mathcal{M} = (\mathcal{S}, \mathcal{A}, p, r)$ be given, where $\mathcal{S}$ is the state space, $\mathcal{A}$ is the action space, $p : \mathcal{S} \times \mathcal{A} \to \Delta(\mathcal{S})$ is the transition probability kernel mapping state-action pairs $(s, a) \in \mathcal{S} \times \mathcal{A}$ to probability distributions $p(\cdot|s, a) \in \Delta(\mathcal{S})$ over the state space, and $r : \mathcal{S} \times \mathcal{A} \to \mathbb{R}$ is the reward function. Given a policy $\pi : \mathcal{S} \to \Delta(\mathcal{A})$ mapping states to probability distributions over the action space, $\mathcal{M}$ evolves as follows: at timestep $t \in \mathbb{N}$, the system is in state $s_t$, action $a_t \sim \pi(\cdot|s_t)$ is selected and executed, a reward $r_t = r(s_t, a_t)$ is received, the state transitions according to $s_{t+1} \sim p(\cdot|s_t, a_t)$, and the process repeats. To each policy $\pi$ is associated the long-run average reward $J(\pi) = \lim_{T \to \infty} \frac{1}{T} \mathbb{E}_\pi[\sum_{i=0}^{T-1} r_i]$, and the goal is to determine an optimal policy $\pi^* = \arg\max_\pi J(\pi)$.

Under mild conditions on the transition kernel and policy (see (Puterman, 2014)), each policy $\pi$ induces a state occupancy measure $d_\pi(\cdot) \in \Delta(\mathcal{S})$ capturing the long-run state visitation behavior induced by $\pi$ over $\mathcal{S}$. When $\mathcal{S}$ is continuous, for a measurable subset $B \subset \mathcal{S}$ we have $d_\pi(B) = \lim_{t \to \infty} P(s_t \in B)$. We will henceforth assume that each measure $d_\pi$ has a corresponding p.d.f. and abuse notation by denoting the value of this p.d.f at $s$ by $d_\pi(s)$. [1] State occupancy measure entropies can be obtained by directly substituting $f = d_\pi$ and $\mathcal{X} = \mathcal{S}$ in the entropy definitions in equation 5, equation 6, and equation 8. In particular, the behavioral entropy induced by $\pi$ resulting from this substitution in equation 8 is given by

$$H^{B,\alpha,\beta}(d_\pi) = \beta \int_{\mathcal{S}} e^{-\beta(-\log(d_\pi(s)))^\alpha} (-\log d_\pi(s))^\alpha ds. \tag{16}$$

We propose to use equation 16 as an exploration objective. To achieve this, we leverage the $k$-NN estimator of equation 13 developed in the preceding section to derive a reward function $r$ such that $J(\pi) \approx H^{B,\alpha,\beta}(d_\pi)$ in the following subsection.

**Behavioral Entropy Reward Derivation.** We next build on the $k$-NN estimator of equation 13 to derive a practical reward function $r$ such that $J(\pi) \approx H^{B,\alpha,\beta}(d_\pi)$. Our derivation is similar to the reward derivations followed in (Liu & Abbeel, 2021; Yarats et al., 2021) for Shannon entropy and (Yuan et al., 2022) for Rényi entropy. Once we are equipped with this reward, we can leverage existing RL methods to learn behavioral entropy-maximizing exploration policies. Let $s_1, \ldots, s_n \sim d_\pi(\cdot)$. Substituting $f = d_\pi$ and $s_i = X_i$ into equation 9, for $i = 1, \ldots, n$, we have that $\widehat{d_\pi}(s_i) =$

---

[1]Similarly, $\pi$ induces a state-action occupancy measure $\lambda_\pi(s, a) = d_\pi(s)\pi(a|s)$. In this work we focus on state occupancy measures, but all results and methods can be extended to apply to state-action occupancy measures in a straightforward manner.

$k\Gamma(d/2+1)/n\pi^{d/2}R_{i,k,n}^d$, where $R_{i,k,n} = \|s_i - NN_k(s_i)\|_2$ and $NN_k(s)$ denotes the $k$-NN of $s_i$ within $\{s_i\}_{i=1,\ldots,n}$. Recalling that $w(x) = e^{-\beta(-\log(x))^\alpha}$, we can write

$$w(\widehat{d}_\pi(s_i)) = e^{-\beta\left(-\left[\log(k\Gamma(d/2+1)) - \log(n\pi^{d/2}R_{i,k,n}^d)\right]\right)^\alpha} \tag{17}$$

$$= e^{-\beta(d\log R_{i,k,n} + D_{k,n})^\alpha}, \tag{18}$$

where $D_{k,n} = -\log(k\Gamma(d/2+1)) + \log(n\pi^{d/2}) = \log\left((n\pi^{d/2})/(k\Gamma(d/2+1))\right)$. Substituting equation 18 into equation 13 gives

$$\widehat{H}_{k,n}^{B,\alpha,\beta}(d_\pi) = -\frac{1}{n}\sum_{i=1}^n \frac{1}{\widehat{d}_\pi(s_i)} w(\widehat{d}_\pi(s_i)) \log w(\widehat{d}_\pi(s_i)) \tag{19}$$

$$= \frac{\beta}{n}\sum_{i=1}^n \frac{n\pi^{d/2}R_{i,k,n}^d}{k\Gamma(d/2+1)} e^{-\beta(d\log R_{i,k,n} + D_{k,n})^\alpha} \left(d\log R_{i,k,n} + D_{k,n}\right)^\alpha \tag{20}$$

$$\propto \frac{1}{n}\sum_{i=1}^n R_{i,k,n}^d e^{-\beta(d\log R_{i,k,n} + D_{k,n})^\alpha} \left(d\log R_{i,k,n} + D_{k,n}\right)^\alpha \tag{21}$$

$$\overset{\propto}{\sim} \frac{1}{n}\sum_{i=1}^n R_{i,k,n}^d e^{-\beta(d\log R_{i,k,n})^\alpha} \left(d\log R_{i,k,n}\right)^\alpha. \tag{22}$$

where the approximate proportionality in equation 22 follows from the fact that, under suitable conditions on $n, k$ (see, e.g., Theorem 1) the contribution of $D_{k,n}$ to the value of equation 21 is negligible. Since $\mathbb{E}_\pi[\widehat{H}_{k,n}^{B,\alpha,\beta}(d_\pi)] \approx H^{B,\alpha,\beta}(d_\pi)$ by Theorems 1 and 2, and since equation 22 is approximately proportional to $\widehat{H}_{k,n}^{B,\alpha,\beta}(d_\pi)$, equation 22 suggests

$$\widetilde{r}(s,a) = \|s - NN_k(s)\|_2^d \, e^{-\beta(d\log\|s-NN_k(s)\|_2)^\alpha} \left(d\log\|s - NN_k(s)\|_2\right)^\alpha \tag{23}$$

as a suitable proxy reward for maximizing behavioral entropy in an RL context. For numerical stability, we follow (Yarats et al., 2021; Liu & Abbeel, 2021) by making the additional simplification of setting $d = 1$ and adding a constant $c > 0$ inside the logarithms to obtain

$$r(s,a) = \|s - NN_k(s)\|_2 \, e^{-\beta(\log(\|s-NN_k(s)\|_2+c))^\alpha} \left(\log(\|s - NN_k(s)\|_2 + c)\right)^\alpha. \tag{24}$$

A visualization of equation 24 with a comparison to the SE reward function is provided in Fig. 13 in the appendix. Armed with this reward, any standard RL method can be applied to learn exploration policies approximately maximizing BE using equation 16. We illustrate its application in data generation for offline RL in the next section. We note that implementing the $k$-NN estimator in equation 24 can be computationally challenging in high dimensions for large $k$ values due to the well-known curse of dimensionality of suffered by $k$-NN estimators Beyer et al. (1999). To address this, in practice $k \leq 15$ is selected and dimension reduction to a feature space of manageable dimensions is performed before $k$-NN estimation is carried out, thereby limiting computational costs Liu & Abbeel (2021); Yarats et al. (2021).

## 5 EXPERIMENTAL RESULTS

The experiments presented in this section (i) provide qualitative insights into the state space coverage achieved by policies that maximize BE, RE, and SE, and (ii) examine the utility of BE-maximizing

| Environment | Task | BE | RE | SE | RND | SMM |
|---|---|---|---|---|---|---|
| Walker | Stand | **990.38** | 988.93 | 954.93 | 947.89 | 496.09 |
| | Walk | **904.66** | 878.20 | 895.89 | 735.77 | 409.46 |
| | Run | 385.07 | **440.53** | 360.64 | 341.03 | 140.29 |
| Quadruped | Walk | **845.31** | 776.64 | 755.79 | 699.22 | 425.11 |
| | Run | **522.32** | 490.75 | 490.46 | 490.66 | 275.38 |

Table 1: Comparison of best offline RL performance across all datasets, training seeds, and offline RL algorithms.

policies for performing offline dataset generation for subsequent offline RL compared with datasets generated using the SE and RE objectives and datasets generated using the RND and SMM algorithms. The state space coverage visualizations that we present suggest that BE-generated

datasets achieve a wider variety of coverage than RE- and SE-generated datasets, and that the RE objective is unstable as a function of $q$ and provides poor coverage for $q > 1$. We provide coverage visualizations for RND and SMM in the appendix. In our offline RL experiments, we demonstrate that offline learning on BE datasets leads to superior performance over SE, RND, and SMM datasets on all five tasks, and superior performance to RE datasets on four out of five tasks (see Table 1).

**Experimental Setup.** For our experiments, we generated BE, RE, SE, RND, and SMM datasets for the Walker and Quadruped environments using the Unsupervised Reinforcement Learning Benchmark (URLB) framework (Laskin et al., 2021)[2]. We subsequently generated t-SNE plots (Hinton & Roweis, 2002) and PHATE plots (Moon et al., 2019) from the BE, RE, SE, RND, and SMM datasets to visualize their varying state space coverage. Finally, we performed offline RL training on all datasets using the Exploratory Data for Offline RL (ExORL) framework (Yarats et al., 2022)[3]. We emphasize that the datasets we generated contained just 500K elements, only 5% as many as the 10M-element datasets considered in the ExORL framework (Yarats et al., 2022), and that we performed just 100K offline training steps, only 20% of the 500K performed in ExORL. Despite these limitations, we achieved comparable performance to that achieved in (Yarats et al., 2022), indicating that using BE-generated data for subsequent offline RL leads to significant improvements in both data- and sample-efficiency.

**Dataset Generation and Visualization.** For dataset generation, we used the Active Pre-Training (APT) algorithm (Liu & Abbeel, 2021) implemented in the URLB framework to maximize BE using the reward proposed in equation 24 for various values of $\alpha$, RE using the reward proposed in Zhang et al. (2021) for various values of $q$, and SE using the default reward from Liu & Abbeel (2021). Specifically, for behavioral and RE we considered $\alpha \in \{0.2, 0.5, 0.7, 0.9, 1.5, 2.0, 3.0, 5.0\}$ and $q \in \{0.2, 0.5, 0.7, 0.9, 1.1, 2.0, 3.0, 5.0\}$. To ensure admissibility of the behavioral entropies we considered, we used the conditioning $\beta = e^{(1-\alpha)\log(\log(M))}$ from Suresh et al. (2024), where $M$ is the dimensions of the representation space. See the discussion following equation 4 in Section 2 for details. For each of the $\alpha$ and $q$ values, as well as for SE, we trained APT on the corresponding reward for 500K pretraining steps on both the Walker and Quadruped environments, collecting the resulting trajectories to form our datasets. This resulted in 17 datasets for each environment, for a total of 34 datasets. To ensure a fair comparison across entropies, for the APT hyperparameters we used the default URLB pretraining hyperparameter values across all datasets (see Table 2 in the appendix). For the datasets generated using the Random Network Distillation (RND) (Burda et al., 2019) and State Marginal Matching (SMM) Lee et al. (2019) algorithms, we similarly trained for 500K pretraining steps and for consistency we used the same RND and SMM hyperparameters considered in URLB.

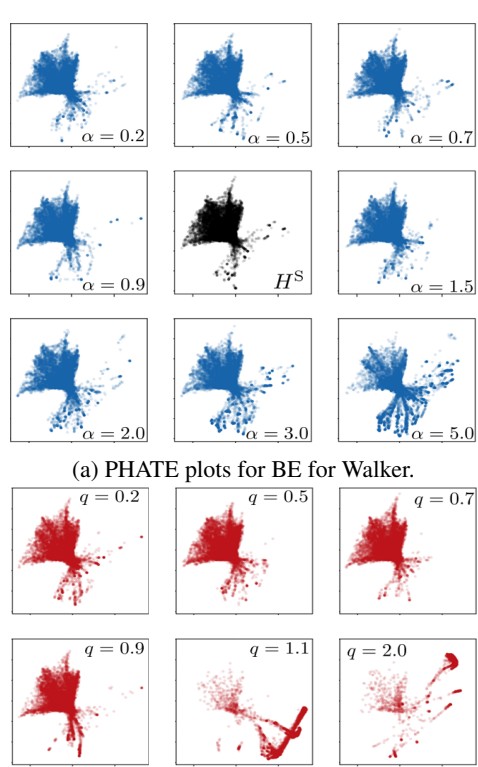

(a) PHATE plots for BE for Walker.

(b) PHATE plots for Renyi for Walker. Coverage for $q = 3.0, 5.0$ similar to $q = 2.0$.

Figure 3: PHATE plots for Walker tasks.

To provide qualitative insight into the state space coverage of the SE, RE, and BE datasets, we generated two-dimensional $t$-SNE (Hinton & Roweis, 2002) and PHATE (Moon et al., 2019) plots of the trajectories they contain.[4] We also generated $t$-SNE and PHATE plots for the RND and SMM

---

[2]https://github.com/rll-research/url_benchmark

[3]https://github.com/denisyarats/exorl

[4]10K representative samples from each dataset were gathered uniformly, totaling 170K samples for each domain from 17 datasets. $t$-SNE and PHATE were implemented on this aggregated 170K-element dataset to ensure uniformity in projections for each dataset and then correspondingly represented individually for clarity.

datasets, pictured in Figure 12 in the appendix. While $t$-SNE has been previously used to visualize RL trajectory data (Zhang et al., 2021), to our knowledge this is the first time PHATE plots have been used to visualize such data. PHATE tends to better retain global structure such as the temporal nature of trajectory data, while $t$-SNE obscures it (Moon et al., 2019). Figure 3 indicates that while both BE and RE-generated datasets provide more flexible levels of state space coverage than SE, BE-generated datasets achieve a wider variety of coverage than RE. See the appendix for $t$-SNE and PHATE plots for the remaining datasets. Importantly, these plots indicate that the RE objective is unstable as a function of $q$ and provides poor coverage for $q > 1$, while the level of coverage provided by BE varies smoothly in $\alpha$. We provide experimental support for this in the following section, where highly unstable offline RL performance on RE datasets and the contrasting stability on BE datasets is illustrated in Figure 4.

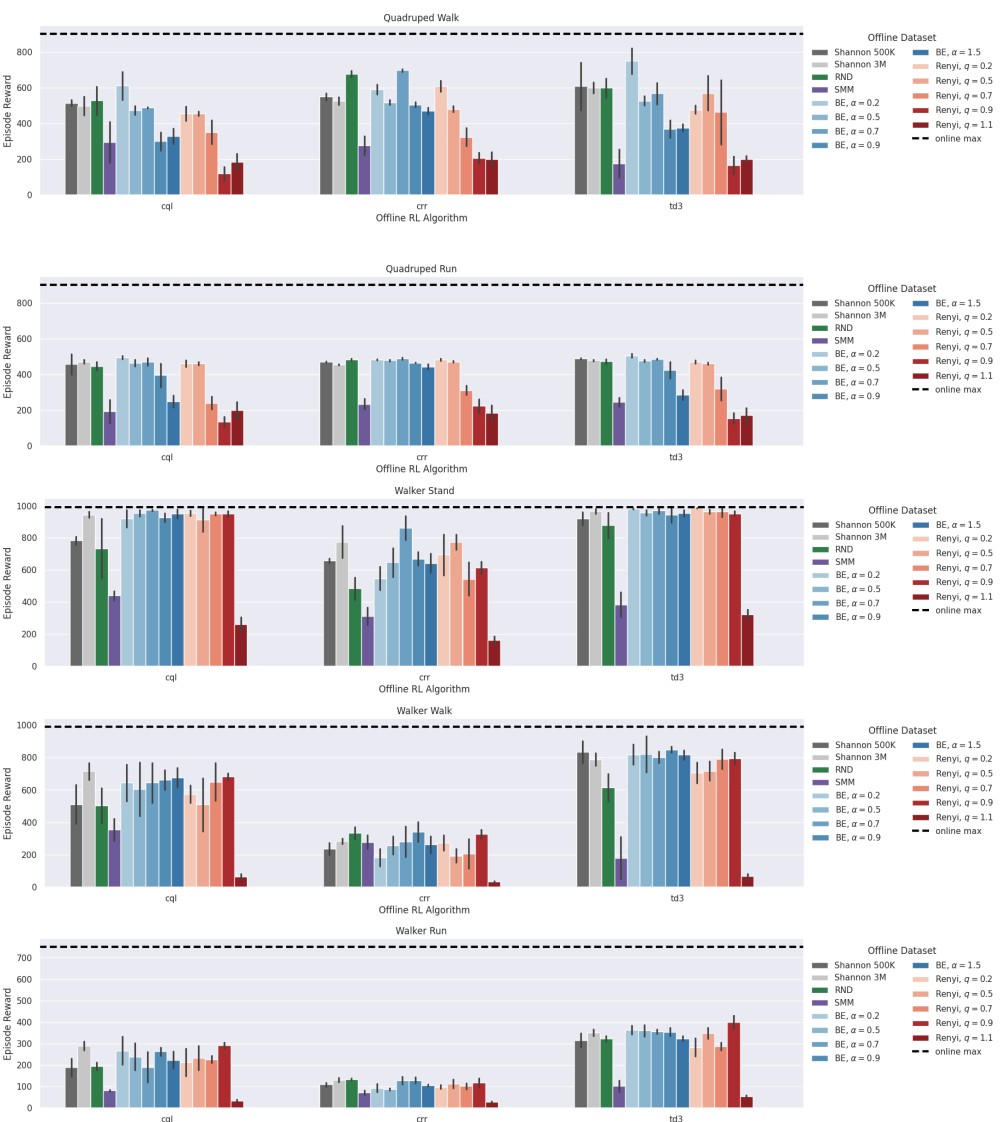

Figure 4: Comparison of offline RL performance over the entropy objectives used in dataset generation. Plots show mean and standard deviation over five seeds. Dotted line shows performance of RL policy trained online until approximate optimality.

**Offline RL Experiments.** We compared the TD3, CQL, and CRR offline RL algorithms (Fujimoto et al., 2018; Kumar et al., 2020; Wang et al., 2020) implemented in the ExORL framework on the datasets generated as described above for all eight $\alpha$ values and for $q \in \{0.2, 0.5, 0.7, 0.9, 1.1\}$. We omitted offline RL training for RE datasets with $q \in \{2.0, 3.0, 5.0\}$ after observing in initial trials that performance was no better than for $q = 1.1$ and typically worse (see poor performance

on $q = 1.1$ datasets in Figure 4). To gain insight into the effect of using larger datasets, we also considered a 3M-element SE-generated dataset. Altogether we considered 17 datasets: eight BE, five BE, two SE, and one each for RND and SMM. On the Walker datasets we considered the Stand, Walk, and Run tasks, while on Quadruped we considered the Walk and Run tasks. For each of the $5 \times 17 = 85$ task-dataset combinations, we trained each of TD3, CQL, and CRR for 100K offline training steps, evaluating performance every 10K training steps. For each of the 255 task-dataset-algorithm combinations, we repeated this training process for a total of 5 different seeds, resulting in our training 1275 offline RL policies altogether. To ensure a fair comparison across all entropies, we used default ExORL hyperparameter values across all datasets (see Table 3 in the appendix).

As summarized in Table 1, offline RL training on BE-generated datasets leads to superior performance over SE-, RND-, and SMM-generated datasets on all five tasks we considered, and superior performance to RE-generated datasets on four out of five tasks. Figure 4 provides a detailed overview of the experimental results for $\alpha \in \{0.2, 0.5, 0.7, 0.9, 1.5\}$ and $q \in \{0.2, 0.5, 0.7, 0.9, 1.1\}$ (complete results for all $\alpha$ values are shown in the appendix). This figure illustrates that BE-generated datasets lead to significantly better performance over the other methods on Quadruped Walk and Walker Walk for the best-performing $\alpha$ values, while offline RL performance on RE datasets for the best-performing values of $q$ is only slightly below that of BE datasets in Quadruped Run and Walker Stand. These trends hold across all algorithms for each of the tasks. On Walker Run, the best-performing RE parameter clearly leads to superior offline RL performance over both BE and SE datasets in the TD3 and CQL trials, but performance on BE datasets is again better than on RE datasets in the CRR trials. Performance on SMM datasets is clearly inferior across all tasks considered. Performance on RND datasets is inferior on Walker tasks, but is almost competitive with BE on Quadruped tasks. Interestingly, the 3M-element SE datasets lead to strong downstream performance on Walker Stand and improved downstream performance over the 500K-element SE datasets on the Walker Stand and Run tasks, but the 3M-element SE datasets actually lead to worse performance compared with the 500K-element SE datasets on both Quadruped tasks and TD3 performance on Walker Walk. Overall, best offline RL performance on BE-generated datasets clearly exceeds best performance on RE datasets on 13 out of 15 task-algorithm combinations and best performance on SE, RND, and SMM datasets on 15 out of 15 task-algorithm combinations.

We observed sensitivity of performance to parameters $\alpha$ and $q$ as well as choice of offline RL algorithm. Regarding the latter, notice in Figure 7 in the appendix that on Walker Walk the SE-generated datasets are competitive with the average and best-performing BE datasets in the TD3 trials, while BE datasets significantly outperform SE ones in both the CQL and CRR trials. On Quadruped Run, on the other hand, performance on BE and SE datasets remains roughly the same across all algorithms, while average RE performance is significantly worse (see appendix for average performance plots for all tasks). These results suggest that offline RL algorithm performance depends in a complex way on the choice of exploration objective used in dataset generation. Well-performing, flexible objectives such as BE – and to a lesser but still significant extent, RE – therefore merit additional study as tools for dataset generation for offline RL.

# 6 CONCLUSION

In this work we developed the theory and practice of behavioral entropy in continuous spaces, enabling the incorporation of human cognitive and perceptual biases into uncertainty perception in complex, real-world scenarios. We first developed and analyzed $k$-nearest neighbor estimators for BE with general probability weightings. We subsequently derived practical reinforcement learning-based methods for maximizing BE under Prelec's probability weighting in sequential decision-making problems, enabling the use of BE as a state space exploration objective in the RL context. Leveraging these algorithmic developments, we experimentally investigated the utility of BE for dataset generation for offline RL. Our experiments demonstrate that BE-generated datasets lead to superior offline RL performance over both Shannon and Rényi entropy-generated datasets, that BE is stabler and therefore easier to use as an exploration objective compared with Rényi entropy, and that BE-generated datasets lead to improved data- and sample-efficiency for offline RL over existing methods. As a limitation, due to the computational burden of dataset generation and offline RL training for a nontrivial variety of $\alpha$ and $q$ values additional environments and offline RL algorithms were not considered. These limitations are important directions for future work.

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

# A  APPENDIX

## A.1  PROOFS

Fix a probability weighting function $w$ and let $g(y) = -\frac{1}{y}\log(w(y))w(y)$. Fix a p.d.f. $f \in \Delta(\mathcal{X})$, where $\mathcal{X} \subset \mathbb{R}^d$ is compact. Fix $n, k \in \mathbb{N}$, and let $X_1, \dots, X_n \sim f(\cdot)$. Recall the definition of $\hat{f}$ from equation 9. Let $\mu$ denote the Lebesgue measure and $B_r(x) = \{x' \in \mathbb{R}^d \mid \|x' - x\|_2 < r\}$. Define

$$H^{B,w}(f) = -\int_{\mathcal{X}} \log(w(f(x)))w(f(x))dx \int_{\mathcal{X}} g(f(x))f(x)dx, \tag{25}$$

$$H_n^{B,w}(f) = -\sum_{i=1}^n \frac{1}{f(x)} \log(w(f(X_i)))w(f(X_i)) = \frac{1}{n}\sum_{i=1}^n g(f(X_i)), \tag{26}$$

$$\widehat{H}_{k,n}^{B,w}(f) = -\sum_{i=1}^n \frac{1}{\hat{f}(x)} \log(w(\hat{f}(X_i)))w(\hat{f}(X_i)) = \frac{1}{n}\sum_{i=1}^n g(\hat{f}(X_i)). \tag{27}$$

Our goal is to establish a bound on the error

$$\left| \mathbb{E}\left[\widehat{H}_{k,n}^{B,w}(f)\right] - H^{B,w}(f) \right|. \tag{28}$$

In general, for finite $k$, even as $n \to \infty$ the approximator $\widehat{H}_{k,n}^{B,w}(f)$ will remain biased due to the biasedness of $\hat{f}$ for fixed $k$ and the lack of a known bias correction procedure for our BE approximator $\widehat{H}_{k,n}^{B,w}(f)$. This contrasts with the situation for simpler estimators like Shannon and Rényi entropies, for which explicit bias correction terms are known (see Singh et al. (2003); Leonenko et al. (2008); Singh & Póczos (2016)). Nonetheless, in Theorem 2 we are able to build on existing results to establish a probabilistic bound on equation 28. We first recall the following result.

**Lemma 1** ((Singh & Póczos, 2016)). *Suppose that, for some $\xi \in (0, 2]$, $f$ is $\xi$-Hölder continuous and strictly positive on $\mathcal{X}$. Suppose furthermore that there exists a function $f_* : \mathcal{X} \to \mathbb{R}^+$ and a constant $f^*$ such that $0 < f_*(x) \le \int_{B_r(x)} f(y)dy/\mu(B_r(x)) \le f^* < \infty$, for all $x \in \mathcal{X}, r \in (0, \sqrt{d}]$, and assume that $\int_0^\infty e^{-x}x^k f(x)dx < \infty$. Then*

$$\left| \mathbb{E}\left[H_n^{B,w}(f)\right] - H^{B,w}(f) \right| = \mathcal{O}\left(\frac{k}{n}\right)^{\frac{\xi}{d}}, \quad (29) \qquad \qquad Var\left(H_n^{B,w}(f)\right) = \mathcal{O}\left(\frac{1}{n}\right). \tag{30}$$

The proof of this result follows directly from that of (Singh & Póczos, 2016, Thm. 5) due to the fact that $H_n^{B,w}(f)$ is an unbiased estimator of $H^{B,w}(f)$. Also note that the variance bound can be trivially strengthened to apply to $\widehat{H}_{k,n}^{B,w}(f)$ due to the fact that the latter is simply the sample average of $n$ i.i.d., bounded random variables:

**Corollary 1.** *Under the conditions of Lemma 1, $Var\left(\widehat{H}_{k,n}^{B,w}(f)\right) = \mathcal{O}\left(\frac{1}{n}\right).$*

It remains to characterize equation 28. We first recall another useful result from the literature. For a given set $S \subset \mathcal{X}$, radius $r$, and $m > 0$, let $\mathcal{N}(S, r)$ denote the covering number, the minimum number of balls of radius $r$ needed to cover $S$. Let $\|\cdot\|_{op}$ denote the operator norm.

**Lemma 2** ((Zhao & Lai, 2022)). *Suppose there exist $C_1, C_2, C_3, \mathcal{N}_0 > 0$ and $\beta \in (0, 1]$ such that the following conditions hold:*

$$(i) \quad \frac{\|\nabla f(x)\|_2}{f(x)} \le C_1; \quad (ii) \quad \frac{\|\nabla^2 f(x)\|_{op}}{f(x)} \le C_2; \quad (iii) \quad \forall t > 0, P(f(x) < t) \le C_3 t^\beta;$$

$$(iv) \quad \mathcal{N}(\{x \mid f(x) > m\}, r) \le \frac{\mathcal{N}_0}{m^\gamma r^d}, \text{ for some } \gamma > 0 \text{ and all } m > 0.$$

*Then, for $\varepsilon > 0$, it holds with probability (w.p.) $1 - \varepsilon$ that*

$$\sup_x \left| \hat{f}(x) - f(x) \right| = \begin{cases} \mathcal{O}\left( \left(\frac{k}{n}\right)^{\frac{2}{d}} \log n + \sqrt{\frac{\log(n/\varepsilon)}{k}} \right) & \text{if } d > 2, \\ \mathcal{O}\left( \frac{k}{n} \log n + \sqrt{\frac{\log(n/\varepsilon)}{k}} \right) & \text{if } d = 1, 2. \end{cases} \tag{31}$$

We are now in a position to prove our main result.

**Theorem 2.** *Suppose $f$ satisfies the conditions of Lemmas 1 and 2. Assume $w$ is Lipschitz continuous. Then, for $\varepsilon > 0$, it holds w.p. $1 - \varepsilon$ that*

$$\left| \mathbb{E}\left[ \widehat{H}_{k,n}^{B,w}(f) \right] - H^{B,w}(f) \right| = \begin{cases} \mathcal{O}\left( \frac{k}{n} \right)^{\frac{\xi}{d}} + \mathcal{O}\left( \left( \frac{k}{n} \right)^{\frac{2}{d}} \log n + \sqrt{\frac{\log(n/\varepsilon)}{k}} \right) & \text{if } d > 2, \\ \mathcal{O}\left( \frac{k}{n} \right)^{\frac{\xi}{d}} + \mathcal{O}\left( \frac{k}{n} \log n + \sqrt{\frac{\log(n/\varepsilon)}{k}} \right) & \text{if } d = 1, 2. \end{cases} \tag{32}$$

*Proof.* First notice that

$$\left| \mathbb{E}\left[ \widehat{H}_{k,n}^{B,w}(f) \right] - H^{B,w}(f) \right| \leq \left| \mathbb{E}\left[ \widehat{H}_{k,n}^{B,w}(f) - H_n^{B,w}(f) \right] \right| + \left| \mathbb{E}\left[ H_n^{B,w}(f) \right] - H^{B,w}(f) \right|. \tag{33}$$

The second term can be bounded using Lemma 1, so it just remains to bound the first term. Recall that $\mathcal{X}$ is compact, $f$ is bounded strictly away from 0 on $\mathcal{X}$, and $w$ is Lipschitz. We therefore have that $g$ is the product of Lipschitz, bounded functions and is therefore itself Lipschitz on its domain. Let $K$ denote the minimal Lipschitz parameter of $g$. Rewriting equation 33 in terms of $g$, we obtain

$$\left| \mathbb{E}\left[ \widehat{H}_{k,n}^{B,w}(f) - H_n^{B,w}(f) \right] \right| = \left| \frac{1}{n} \sum_{i=1}^{n} \mathbb{E}\left[ g(\hat{f}(X_i)) - g(f(X_i)) \right] \right| \tag{34}$$

$$\leq \frac{1}{n} \sum_{i=1}^{n} \mathbb{E}\left[ \left| g(\hat{f}(X_i)) - g(f(X_i)) \right| \right] \tag{35}$$

$$\overset{(a)}{=} \mathbb{E}\left[ \left| g(\hat{f}(X_1)) - g(f(X_1)) \right| \right] \tag{36}$$

$$\leq K\mathbb{E}\left[ \left| \hat{f}(X_1) - f(X_1) \right| \right] \tag{37}$$

$$\leq K\mathbb{E}\left[ \sup_x \left| \hat{f}(x) - f(x) \right| \right] \tag{38}$$

where equation a follows from the fact that the $X_1, \ldots, X_n$ are i.i.d. An application of the law of total probability and Lemma 2 to the last term completes the proof. $\qquad\square$

## A.2 HYPERPARAMETERS

| Optimization hyperparameter | Value |
|---|---|
| replay buffer capacity | $10^6$ |
| mini-batch size | 1024 |
| agent update frequency | 2 |
| discount factor ($\gamma$) | 0.99 |
| optimizer | Adam |
| learning rate | $10^{-4}$ |
| critic target rate ($\tau$) | 0.01 |
| exploration stddev clip | 0.3 |
| exploration stddev value | 0.2 |
| APT hyperparameter | |
| forward net architecture | $(512 + \dim(\mathcal{A})) \to 1024 \to 512$ ReLU MLP |
| inverse net architecture | $(2 \times 512) \to 1024 \to \dim(\mathcal{A})$ ReLU MLP |
| representation dimension | 512 |
| $k$ in NN approximator | 12 |
| average top $k$ in NN | True |
| RND hyperparameter | |
| representation dimension | 512 |
| predictor, target network architecture | $\dim(\mathcal{S}) \to 1024 \to 1024 \to 512$ ReLU MLP |
| normalized observation clipping | 5 |
| SMM hyperparameter | |
| skill dimension | 4 |
| skill discriminator learning rate | $10^{-3}$ |
| VAE learning rate | $10^{-2}$ |

Table 2: Data generation hyperparameters

| Shared hyperparameter | Value |
|---|---|
| replay buffer capacity | $10^6$ |
| mini-batch size | 1024 |
| agent update frequency | 2 |
| discount factor ($\gamma$) | 0.99 |
| optimizer | Adam |
| number of hidden layers | 2 |
| hidden dimension | 1024 |
| learning rate | $10^{-4}$ |
| critic target rate ($\tau$) | 0.01 |
| training steps | $10^5$ |
| TD3 hyperparameter | |
| stddev clip | 0.3 |
| CQL hyperparameter | |
| CQL-specific $\alpha$ | 0.01 |
| Lagrange | False |
| number of sample actions | 3 |
| CRR hyperparameter | |
| number of samples to estimate $V$ | 10 |
| transformation | indicator |

Table 3: Offline RL hyperparameters

## A.3 ADDITIONAL OFFLINE RL EXPERIMENTS

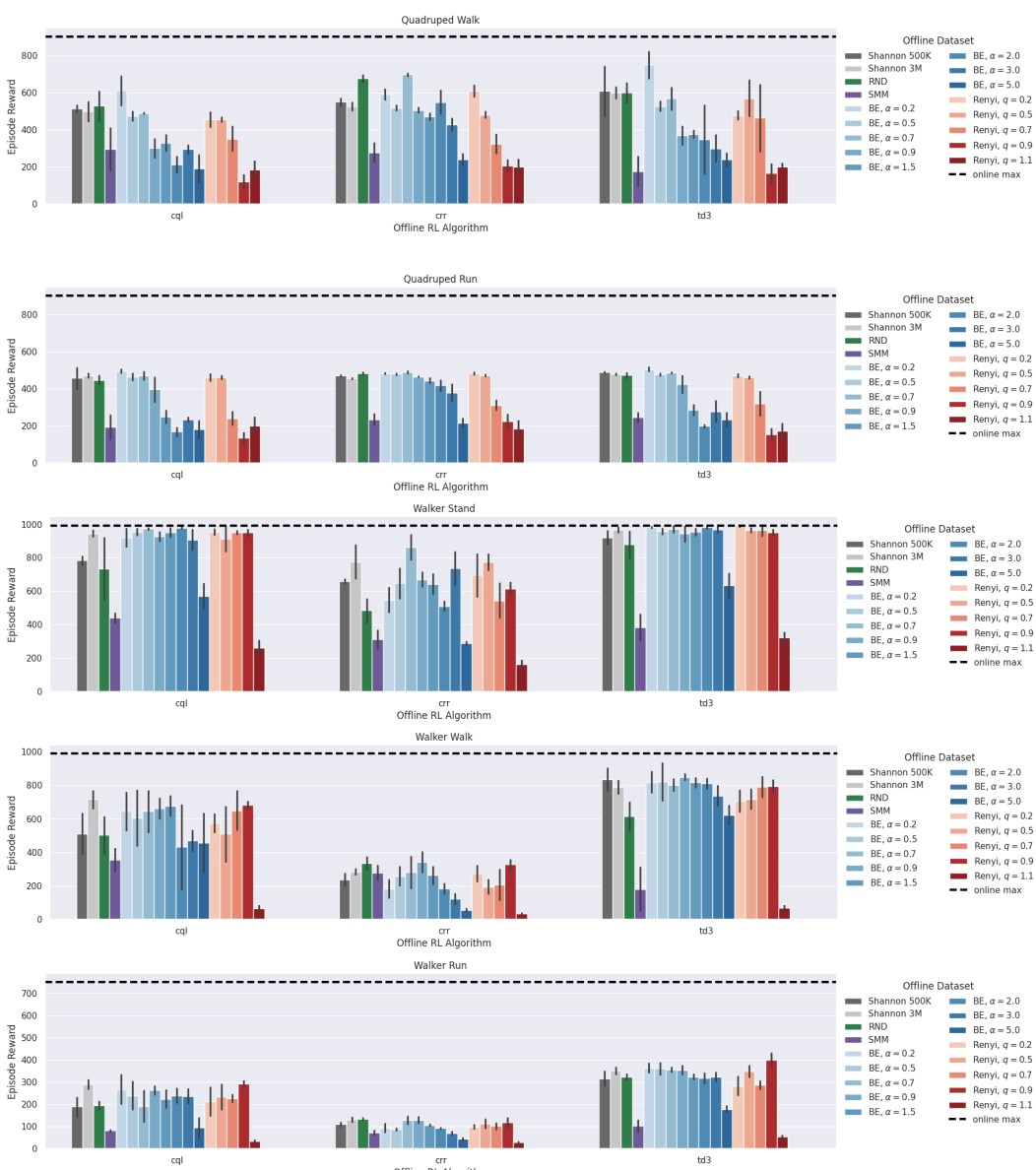

Figure 5: Offline RL results for all $\alpha$ and $q$ values evaluated. Initial trials showed $q \in \{2.0, 3.0, 5.0\}$ led to performance no better (and usually worse) than $q = 1.1$, so offline RL training for these $q$ values was not performed.

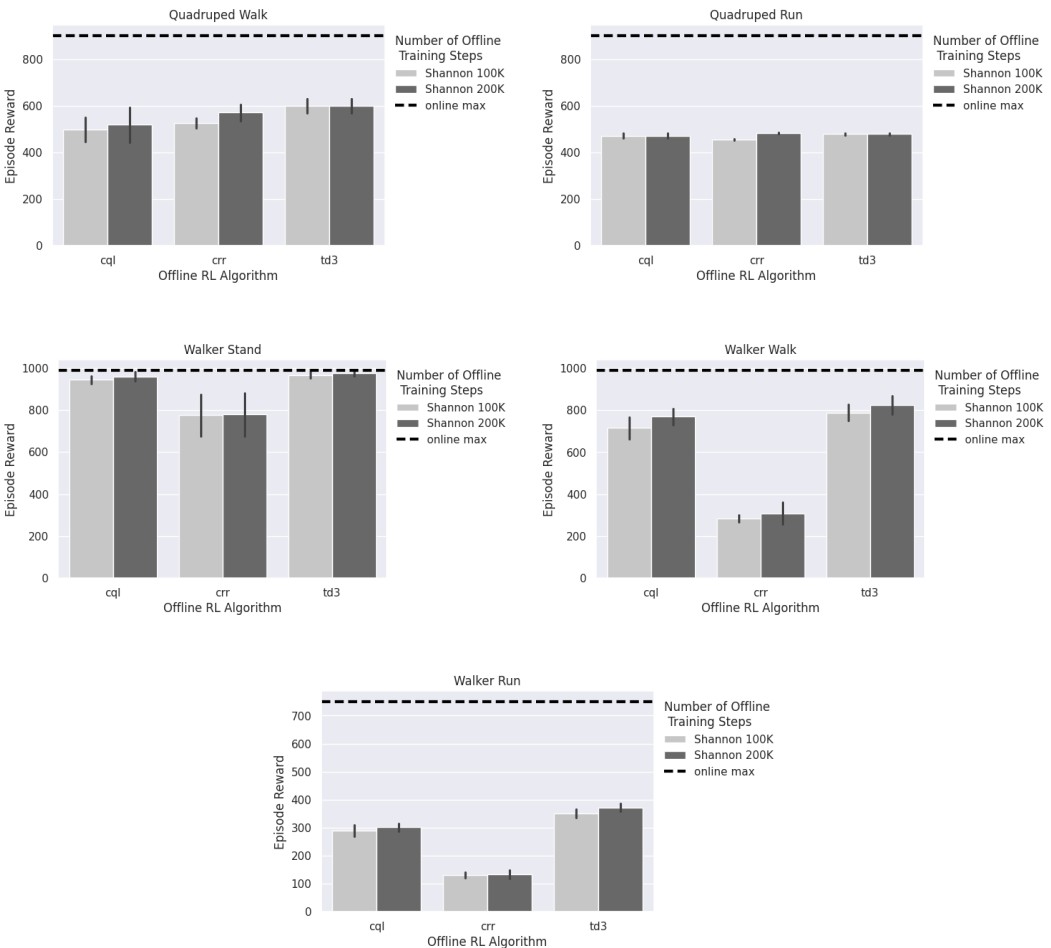

Figure 6: Ablation result comparing the effect of performing 100K vs. 200K offline RL training steps on a 3M-element dataset generated using Shannon entropy as exploration objective. These results suggest that performing additional offline RL training has only a marginal effect on downstream task performance.

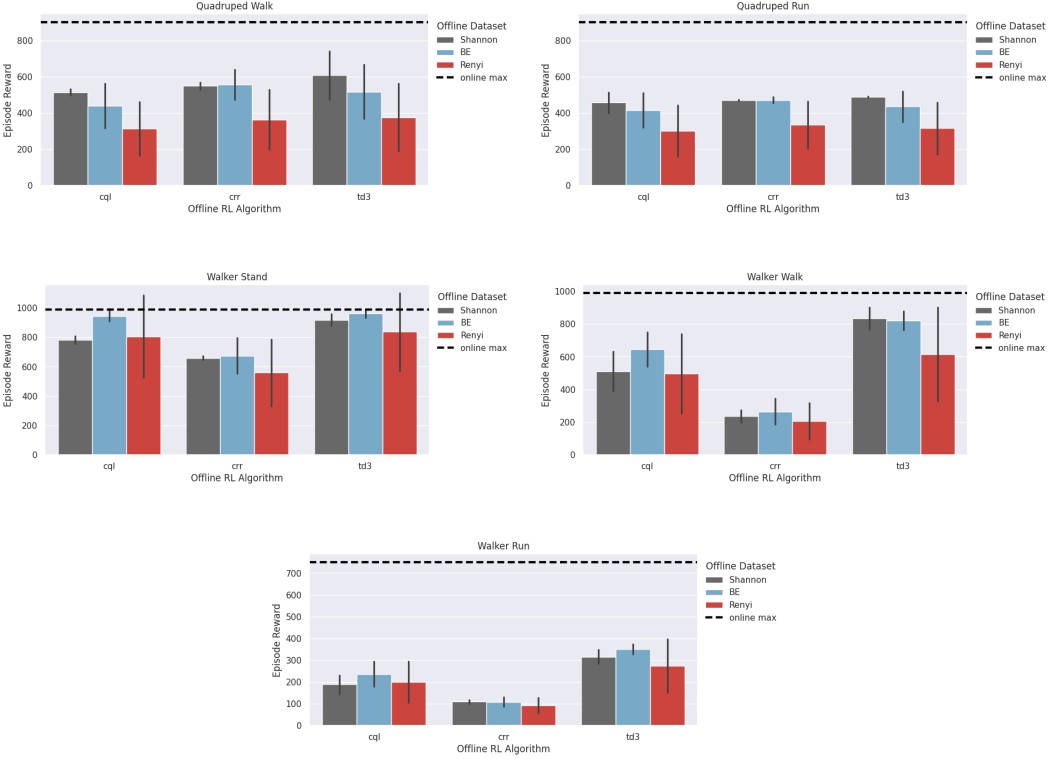

Figure 7: Offline RL results averaged over all $\alpha, q$ values.

## A.4 QUANTITATIVE COVERAGE EXPERIMENTS

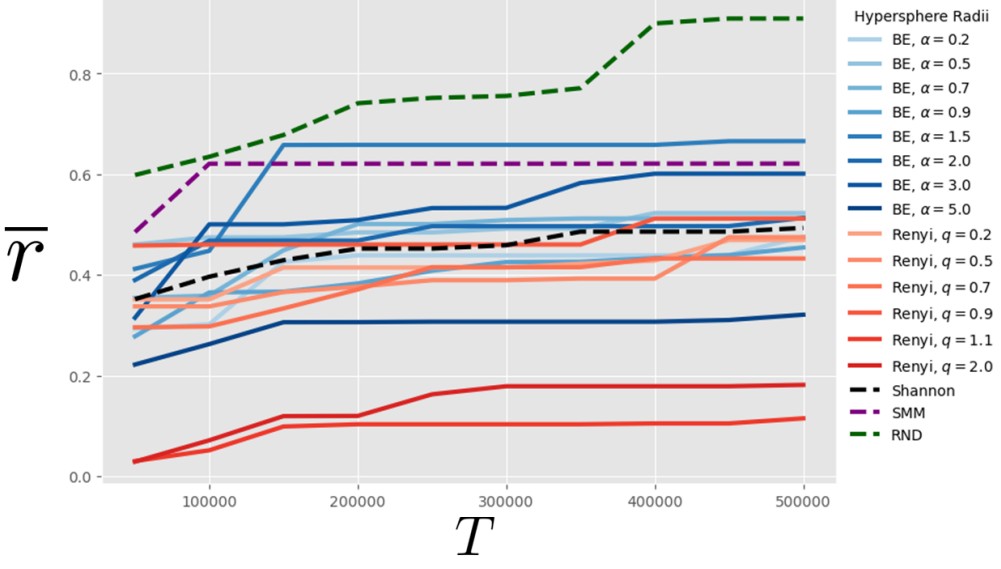

(a) Volumetric coverage for data generation on Walker

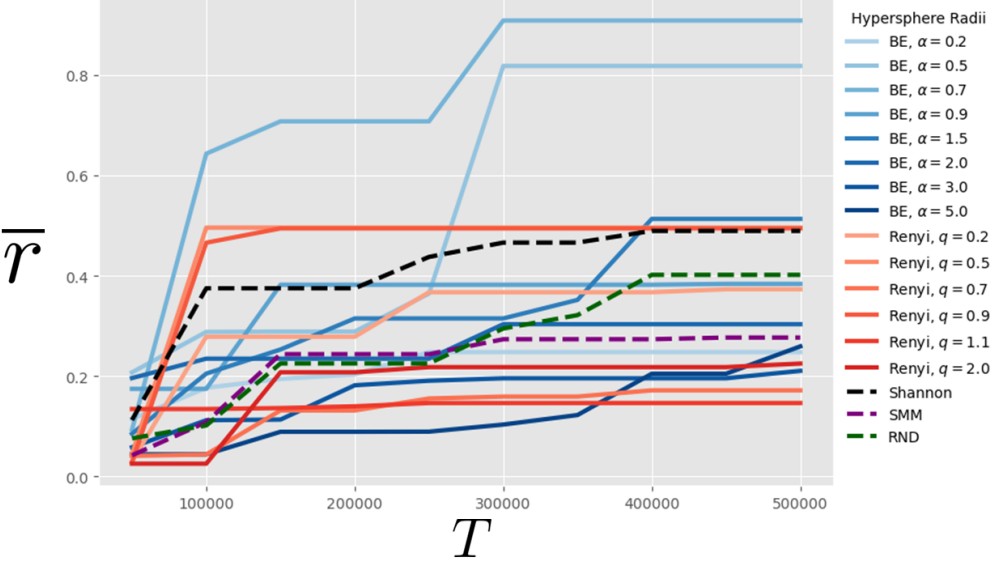

(b) Volumetric coverage for data generation on Quadruped

Figure 8: Visualization of evolution of smallest hypersphere radius $\overline{r}$ (normalized by the maximum radius achieved over all datasets) over the course of data generation training step $T$ for the Walker and Quadruped domains. We refer to this coverage metric as *volumetric coverage*. Welzl's algorithm was used to determine the radius $\overline{r}$. 10K data points were sampled uniformly from every 50K iteration increment and cumulatively added to get a total of 100K samples for 500K iterations. Volumetric coverage varies considerably with the choice of parameters and data generation methods. For the range of parameters $\alpha$ and $q$ that we considered, BE exhibits higher volumetric coverage than RE on average on the problems under consideration. SE volumetric coverage was about average, while RND and SMM volumetric coverage differed sharply across domains: RND outperformed all other data generation methods on Walker, while SMM was not far behind; on Quadruped, on the other hand, both underperformed. Values of $q < 1$ enjoy higher volumetric coverage for RE on both tasks, while values of $\alpha < 1$ enjoy higher volumetric coverage for BE on Quadruped; since $q < 1, \alpha < 1$ tend to correspond to superior downstream offline RL performance (see Fig. 5), this suggests volumetric may be positively correlated with performance on downstream tasks. We also note note that RE with $q > 1$ in general shows both poor volumetric coverage and poor qualitative coverage (PHATE and t-SNE), which might correspond to its poor performance in all tasks. These relationships are not conclusive, however, and further investigation is needed.

## A.5 ADDITIONAL QUALITATIVE VISUALIZATIONS

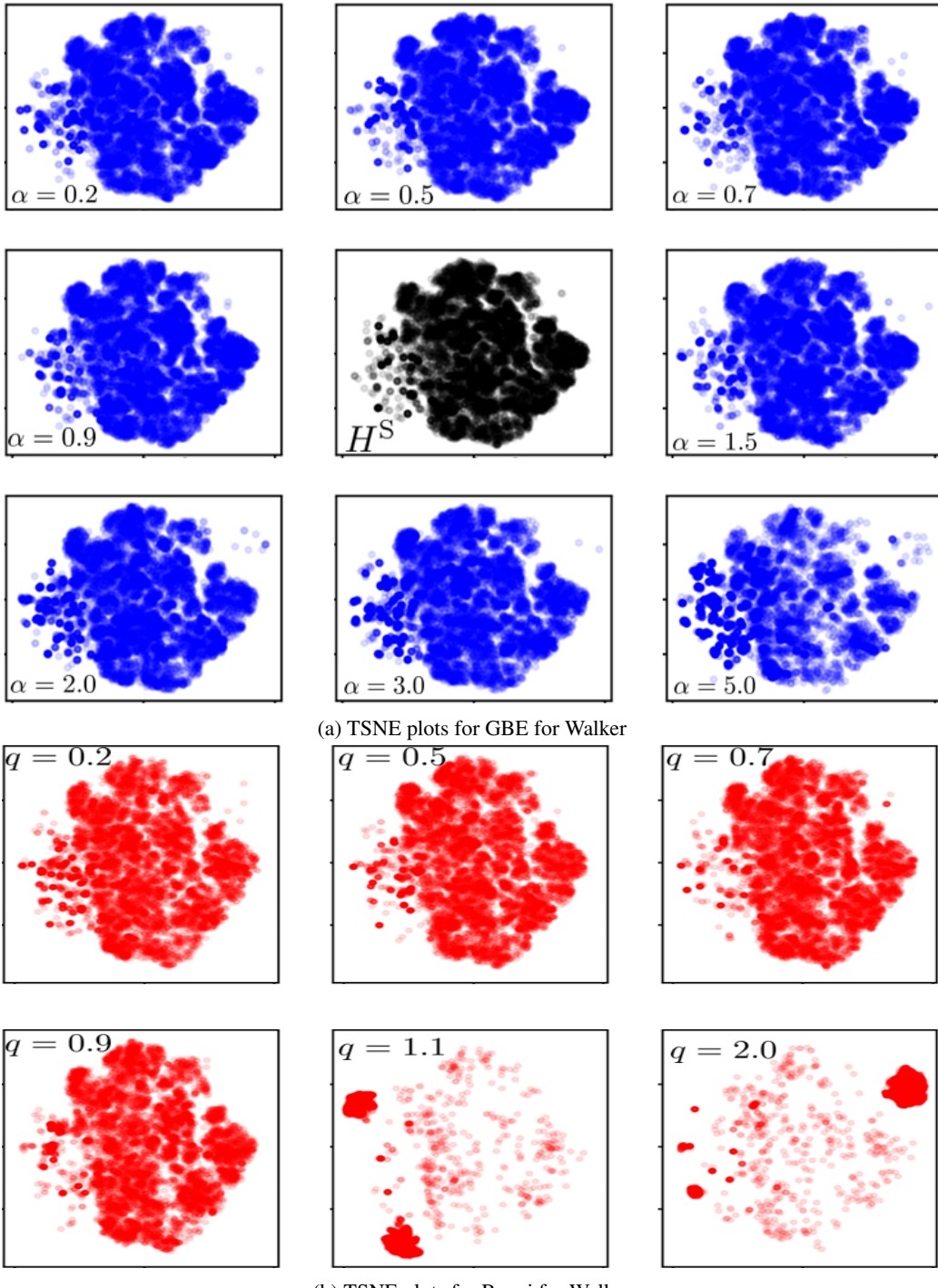

(a) TSNE plots for GBE for Walker

(b) TSNE plots for Renyi for Walker

Figure 9: TSNE plots for Walker

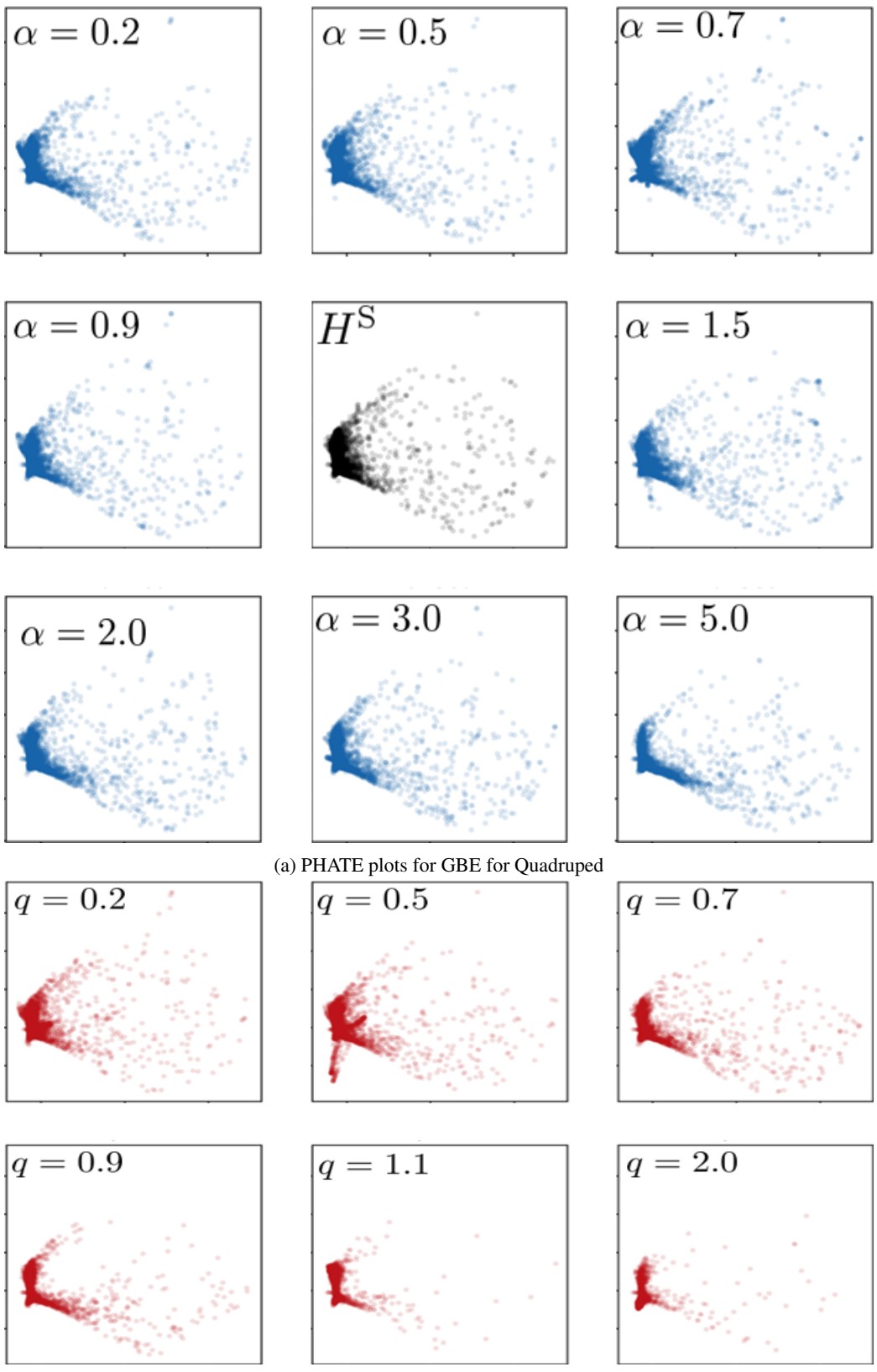

(a) PHATE plots for GBE for Quadruped

(b) PHATE plots for Renyi for Quadruped

Figure 10: PHATE plots for Quadruped

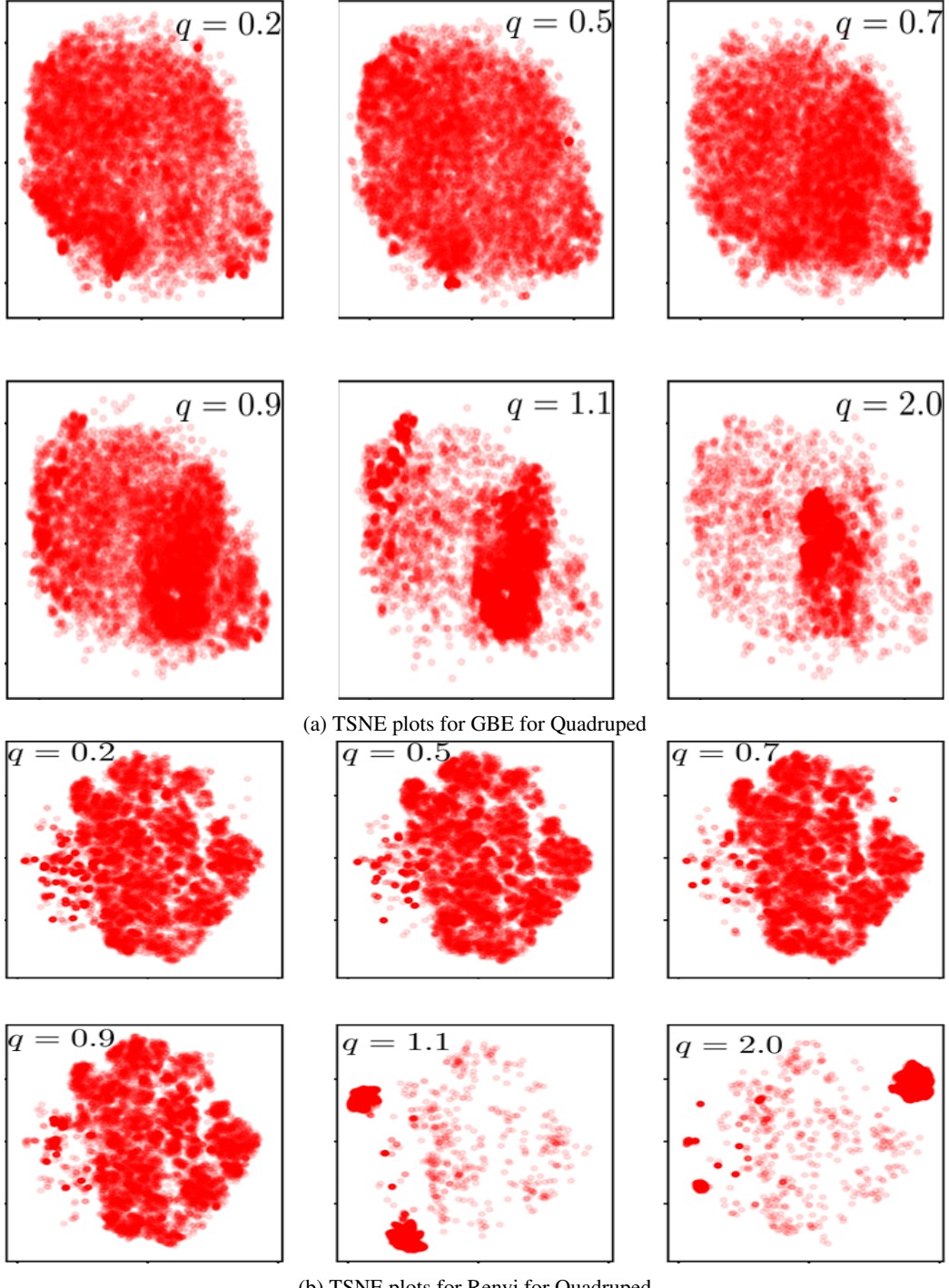

(a) TSNE plots for GBE for Quadruped

(b) TSNE plots for Renyi for Quadruped

Figure 11: TSNE plots for Quadruped

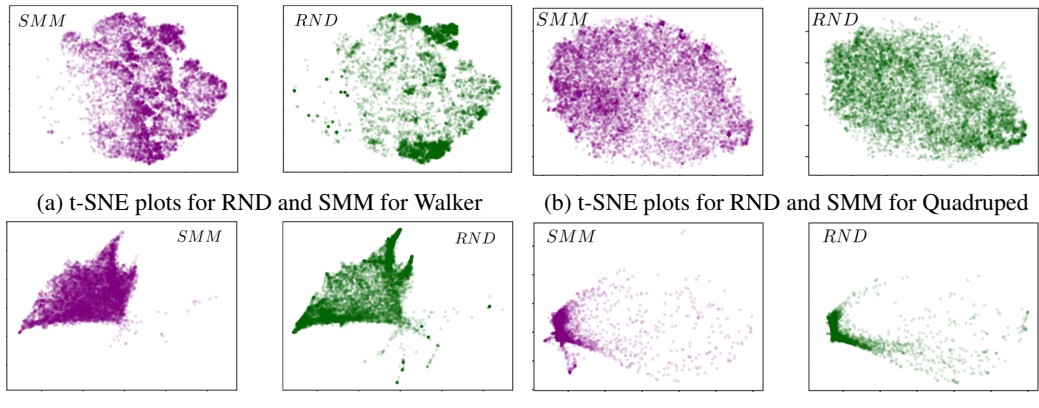

(a) t-SNE plots for RND and SMM for Walker  (b) t-SNE plots for RND and SMM for Quadruped

(c) PHATE plots for RND and SMM for Walker  (d) PHATE plots for RND and SMM for Quadruped

Figure 12: Qualitative visualization of SMM and RND for data generation

## A.6  BE Reward Function Visualization

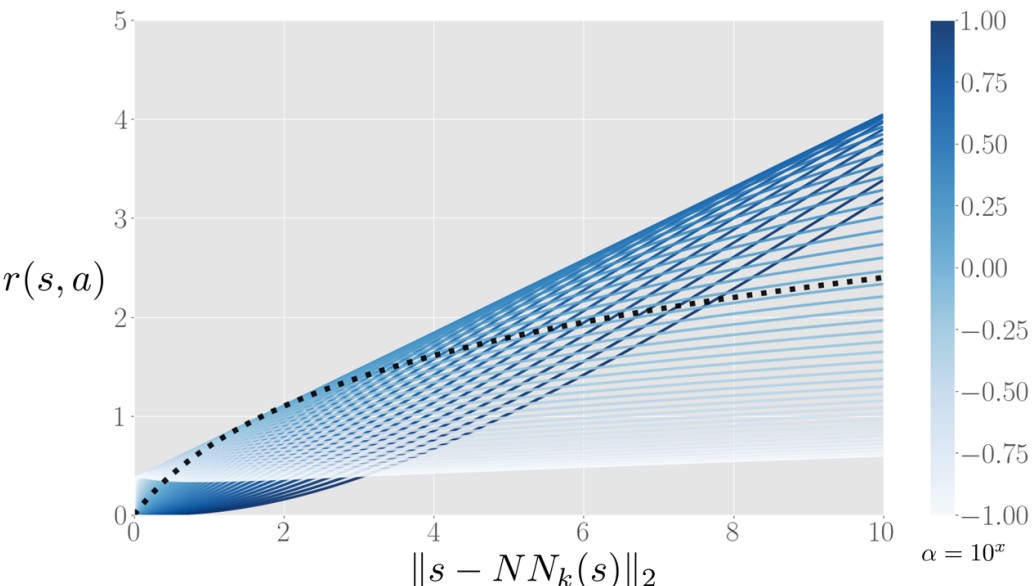

Figure 13: Visualization of the BE reward function equation 24 by varying the parameter $\alpha$ with $\beta$ conditioned according to (4) from (Suresh et al., 2024) with $M = 512$, denoting the representation dimensions. These visualizations highlight the diversity and variety of rewards that can be obtained by a BE-maximizing reward function (blue region) as compared to the single SE objective (dotted black line).

