# OpenReview forum: "Behavioral Entropy-Guided Dataset Generation for Offline Reinforcement Learning"
_ICLR.cc/2025/Conference — ICLR 2025 Poster_

### Official Review · Reviewer_T8Fn · 2024-11-02

**Soundness:** 3
**Presentation:** 3
**Contribution:** 2
**Rating:** 6
**Confidence:** 4

**Summary:**

The paper proposes to use behavioral entropy (BE), a generalized version of Shannon's entropy using a reweighting function, as reward to generate datasets and tests whether these datasets would be better suited for offline RL.
The behavioral entropy allows reweighting low-probable events and high-probable events, and it was used in Economics to explain human behavior.
The paper shows that data generated with maximizing BE can lead to higher downstream task performance in offline RL.

**Strengths:**

- analysis of behavior entropy on exploration
- nice visualizations
- continuous space estimators for BE

**Weaknesses:**

- another hyperparemeter introduced, not unsurprising that better performance can be achieved by tuning this parameter per environment
- computationally heavy (because of k-NN that needs to be rebuild every time) and no mentioning of that
- no comparison to simple baselines like RND
- no improvements for strong offline methods

**Questions:**

- The intuition behind BE did not came across well. Also, why would one want a different weighting of high and low-prob. values?
- The graphics look nice, but in the context of beh. generation and exploration, there is no parametric underlying distribution (like Bernoulli) but essentially, one is trying to achieve uniform distribution over the state-space. An illustration on how this exploration strategy really performs differently and how would be interesting.

- how often does the $k$-NN tree for fast retrieval be rebuild? After every episode?
- Since alpha<1 and q<1 work best, one could say that seldomly visited states (low probability) should give a higher weighting in the reward / entropy
- How was alpha / q selected?
- How does the method scale to high-dimensional input?

## Post rebuttal update
the authors have improved the paper significantly and addressed most of my concerns. I am increasing my score to 6.

---

> ### Author Response · Authors · 2024-11-23
> **Response to Reviewer T8Fn -- Part 1**
>
> We thank the reviewer for devoting their time and effort to reading and evaluating our manuscript. We appreciate the comments and feedback provided. We are pleased that the reviewer appreciated our extension of BE to continuous spaces, development of $k$-NN BE estimators, and our visualizations. We have thoroughly responded to each of the reviewer's concerns and questions below.
>
> ---
>
> > **Weakness 1:** Another hyperparemeter introduced, not unsurprising that better performance can be achieved by tuning this parameter per environment
>
> **Response to Weakness 1:**
>
> We are puzzled by this statement, which seems to oversimplify our contribution and underestimate the difficulties involved in developing principled RL-based methods for exploration. Is the reviewer suggesting that investigating families of entropies beyond SE for data generation in offline RL is not worthwhile? Any clarification the reviewer can provide on this point will be much appreciated.
>
> In any case, we respectfully reaffirm the significance of our contribution, which we again summarize here. In this work we considered a parametrized class of entropies (BE parametrized by $\alpha$), but before they could be used as practical exploration objectives we first had to extend the definition of BE to continuous spaces, develop and analyze $k$-NN estimators for BE, and derive corresponding RL-amenable reward functions. These are significant theoretical contributions. Once this had been achieved, we experimentally evaluated the effect of using BE as a data generation method for offline RL and observed that the wide range of coverages provided by BE datasets led to superior performance compared with existing methods. These results together establish the usefulness and importance of BE as a data generation method for offline RL.
>
>
> ---
>
> > **Weakness 2:** computationally heavy (because of k-NN that needs to be rebuild every time) and no mentioning of that
>
> **Response to Weakness 2:**
>
> We thank the reviewer for raising this important concern. We want to clarify that the $k$-NN computation is carried out on each mini-batch sampled from the replay buffer. This is in keeping with the APT and Proto-RL [Yarats et al., 2021] algorithms, and, as discussed in Section 5 starting around line 386 of our submission, we used APT to maximize behavioral entropy (BE), Renyi entropy (RE), and Shannon entropy (SE). Since mini-batch sizes were fixed at $1024$ (see Table 2 in the appendix), the **computational cost of computing $k$-NNs for $k = 12$ is small**.
>
> ---
>
> > **Weakness 3:** no comparison to simple baselines like RND
>
> **Response to Weakness 3:**
>
> Thanks to the reviewer for raising this valid point. To address this concern **we have performed additional offline RL experiments on both RND- and SMM-generated datasets** for the rebuttal and have added these to the revision (see Figs. 4, 5). As summarized in Table 1 in the revision, performance on BE datasets was superior to performance on SE, RND, and SMM datasets across all tasks. We hope that these additional results satisfy the reviewer, but we also emphasize that our original submission did compare our BE-maximizing approach with the popular SE-maximizing approach of APT (the specific algorithm we compare against is the ICM-based implementation of APT, as described in Section 5) as well as an RE-maximizing variant of this algorithm. We believe that the revised experiments strengthen the contribution of our paper and again thank the reviewer for suggesting this improvement.
>
> ---
>
> > **Weakness 4:** no improvements for strong offline methods
>
> **Response to Weakness 4:**
>
> Could the reviewer please clarify what they mean by "improvements for strong offline methods"? We emphasize that the scope of our submission is the application of BE to **data generation for offline RL**. We do not claim to improve existing offline RL methods or to devise a new offline RL method. This is definitely an important question, but is out of scope for the present submission.

---

> ### Author Response · Authors · 2024-11-23
> **Response to T8Fn -- Part 2**
>
> > **Question 1:** The intuition behind BE did not came across well. Also, why would one want a different weighting of high and low-prob. values?
>
> **Response to Question 1:**
> As illustrated in Fig. 1, different probability weightings induce a variety of entropy valuations. When BE is used as an exploration objective, this fact will yield a variety of policies. On one end of the spectrum, when BE valuation is low an exploration policy tends to ignore low probability areas and focuses instead on refining the coverage of high probability areas, thereby creating denser and more detailed coverage. On the other end of the spectrum, where BE valuation is high an exploration policy gets attracted to low probability areas and focuses on refining the coverage of low probability areas, thus creating sparser and wider coverage. **We have modified the introduction** in the revision to clarify the motivation for using BE for data generation. **We have also included a new visualization** and additional discussion of the BE reward function from eq. (24) in the revision (see Fig. 13) to provide further insight into how BE differs from SE and how it might be expected to affect coverage.
>
>
> ---
>
> > **Question 2:** The graphics look nice, but in the context of beh. generation and exploration, there is no parametric underlying distribution (like Bernoulli) but essentially, one is trying to achieve uniform distribution over the state-space. An illustration on how this exploration strategy really performs differently and how would be interesting.
>
> **Response to Question 2:**
>
> We emphasize to the reviewer that the qualitative coverage results in Section 5 and the appendix (see Figs. 3, 9, 10, 11, 12) provide extensive visualizations using both $t$-SNE and PHATE plots of the coverage provided on both Walker and Quadruped by all of the data generation algorithms considered in the paper. Specifically, **these visualizations illustrate the diverse kinds of coverage provided by BE for different values of $\alpha$ and how this compares with the other methods.** As mentioned above, we have also included a new visualization of how our proposed BE reward (eq. (24)) varies with $\alpha$ as well as how it compares to the standard SE reward (see Fig. 13). This figure illustrates that BE produces a diverse set of rewards that can both under-weigh and over-weigh the SE reward (dotted black line), inducing a variety of exploration strategies depending on choice of $\alpha$.
>
>
> ---
> > **Question 3:** how often does the k-NN tree for fast retrieval be rebuild? After every episode?
>
> **Response to Question 3:**
>
> The $k$-NN computation is carried out on a per-batch basis. Please refer to our response to **Weakness 2** for additional details.
>
> ---
> > **Question 4:** Since alpha<1 and q<1 work best, one could say that seldomly visited states (low probability) should give a higher weighting in the reward / entropy
>
> > **Question 5:** How was alpha / q selected?
>
> **Response to Questions 4 and 5:**
>
> The question of how to select the best value of $\alpha$ (for BE) or $q$ (for RE) for optimal data generation is a good one. For our experiments we have simply considered an evenly distributed range of $\alpha$ and $q$ values and, as the reviewer observed, the values $\alpha, q < 1$ lead to superior offline RL performance on downstream tasks. As discussed in our response to Question 2, some intuition about the effect of parameter choices is provided by Fig. 13 provided in the revision, but without prior information about the domain and the specific downstream tasks to be solved, it is currently unclear how $\alpha$ and $q$ might be selected more systematically. Nonetheless, we believe that with either prior information about the domain and downstream tasks or feedback during the unsupervised RL portion of training, it should be possible to adapt the values of $\alpha$ and $q$ during data generation to provide superior coverage for learning the downstream tasks at hand. This is an interesting direction for future work.
>
> ---
>
> > **Question 6:** How does the method scale to high-dimensional input?
>
> **Response to Question 6:**
>
> Our method scales well to high-dimensional state spaces due to its use of APT as its base algorithm. As discussed in our submission beginning around line 384, we used the APT algorithm as our base method for maximizing BE, RE, and SE. One of the key features of the APT method is that it learns a feature mapping into a feature space of *user-defined dimension*, then performs SE maximization using **$k$-NN estimation over the resulting feature space.** This allows the APT method to handle potentially high-dimensional state spaces via user selection of an appropriate feature space dimension and, since we use APT as our base algorithm, we enjoy this feature as well.

---

> > ### Comment · Reviewer_T8Fn · 2024-11-26
> > **Acknowledging author's response**
> >
> > Thank you for your responses.
> >
> > Weakness 1: Another hyperparameter:
> > Whenever you introduce an additional parameter that is specific to the environment, you can expect a gain from tuning it. Independently on whether it is via BE, SE or a completely different method. What does it practically mean? I have to run many data-collection runs with different alphas, then train an offline agent and then select. Doesn't that seem odd? In offline RL we want to learn from existing data as good as possible. Tweaking the data / changing the way it is collected to obtain better performance in offline RL is only reasonable if that transfers to a new setting.
> >
> > W2: computation
> > okay. thanks for the clarification. Still it was and is not written in the paper. It only refers to existing methods and the reader would need to read those papers to understand a basic detail. As a remark: Computing entropies in high-dim with 1024 points is quite rough. But I understand that this can work in practice good enough.
> >
> > W3: thanks for the comparison. However, the there are also hyperparameters that would need to be adapted as you did it for your own method.
> >
> > W4: strong offline methods. I meant that when using TD3, which gives best results, there is very little difference between SE and BE.
> >
> > Thanks for answering my questions as well.
> > I acknowledge the improvement of the paper. However, I still have reservations about the contribution of this paper to the community and will keep my evaluation score for now.

---

> > > ### Author Response · Authors · 2024-11-30
> > >
> > > Thanks for the response. Our responses follow:
> > > 1. In this paper we study dataset generation for subsequent offline RL, where **we have control over how the datasets are generated**. We are not proposing a new offline RL method, where "we want to learn **from existing data** as good as possible". The goal in our setting is to generate datasets that lead to good offline RL performance on a variety of tasks. As our experiments demonstrate, using BE as an exploration objective leads to such datasets. We emphasize that we are not proposing a new problem, as dataset generation within unsupervised and offline RL has been widely studied (see references in intro).
> > > 2. We are glad our response regarding the $k$-NN computation satisfied the reviewer. We will include a brief description in the final revision.
> > > 3. Can the reviewer please clarify what is meant by "the there are also hyperparameters that would need to be adapted as you did it for your own method"? Note that we used the same RND and SMM hyperparameters as used in URLB and ExORL (see Table 3 in the revision), and that no hyperparameter tuning of the base unsupervised RL algorithms was performed in our experiments.
> > > 4. We respectfully disagree with the reviewer that "there is very little difference between SE and BE" when TD3 (or CQL or CRR) is used. As summarized on lines 511-514 of our submission, in our experiments "best offline RL performance on BE-generated datasets clearly exceeds best performance on RE datasets on 13 out of 15 task-algorithm combinations and best performance on SE, RND, and SMM datasets on 15 out of 15 task-algorithm combinations." Given that offline RL training for each task-algorithm combination was evaluated over 5 independent replications, we emphasize that **the superior offline RL performance observed on BE-generated datasets is statistically meaningful**, including for the TD3 results. Note that in our revision we have shown that this trend continues to hold for additional dataset generation methods (see Figures 4 and 5 in the revision).

---

> ### Author Response · Authors · 2024-11-25
>
> Thanks again for your time and effort in reviewing our paper. We are following up to ensure that our response and revision have addressed all your concerns. We are happy to address any remaining questions before the end of the discussion period tomorrow.

---

> ### Author Response · Authors · 2024-12-02
>
> We are following up to ensure that our response to your last round of questions has addressed all your concerns. Please let us know if there are any final questions before the end of the discussion period today.

---

### Official Review · Reviewer_B32C · 2024-11-02

**Soundness:** 2
**Presentation:** 2
**Contribution:** 3
**Rating:** 6
**Confidence:** 4

**Summary:**

Entropy-based objectives are widely used to perform state space exploration in reinforcement learning (RL) and dataset generation for offline RL. In this paper, the authors borrow the idea of behavioral entropy (BE) from a prior robotics exploration paper and extend it to continuous space. This paper argues that, compared to prior entropy metrics, using BE as the RL objective enables more complete state coverage.

**Strengths:**

State coverage is an important direction of unsupervised RL, which could benefit downstream task solving by learning useful skills, or, as in this paper, by generating a dataset for offline RL.

Regarding originality, even though the idea of applying BE to exploration originates from a cited prior work, to the best of my knowledge, the idea of applying it to state coverage is novel. Further, the authors extend BE from discrete to continuous space and propose a knn-based measurement.

Regarding clarity, the figures in the paper are well designed: Fig 2 clearly illustrates the property of BE under different values of $\alpha$ (one of its parameter) and the differences between BE and prior entropy metrics. Meanwhile, the paper uses the same color for each entropy metric across the paper, which further eases the reading.

**Weaknesses:**

My main questions are about the evalution on state coverage and offline RL performance. Specifically:
1. Could authors explain, in Fig3, why visualizing each experiment using separate plots is a valid metric of state coverage? IIUC, PHATE, t-SNE, and methods does some non-linear projection to project data to 2D, and this brings two problems when comparing these figures:
* As state trajectories go through non-linear projection, the distance between points more reflect their similarity rather their true distance, thus making it hard to say more spanned points covers large state space (I may be wrong on this). Regarding this, one way may be to visualize trajectories of some points and see if they are indeed very distinct states. Another way I would like to see is to run experiments on some 2d domains so we can visualize without PHATE.
* As each experiment is visualized separately, the PHATE may use different projections in different subfigures, making it hard to say points indeed spread out or the projection makes them look spread out. Regarding this, one more fair way is to draw PHATE plots on trajectories across all experiments to ensure they share the same projection.
2. If we assume PHATE plot is a valid, I still have some questions:
* I do not see singnificant difference between SE and BE for Quadruped coverage, is there any? If not, why that's the case?
* In offline RL experiments, do you train each method until it converges? Authors mentions "we performed just 100K offline training steps", but I thought it's more fair to compare performance under convergence.
* The authors motivates in the intro that better state coverage should lead to better offline RL performance, but in Fig 6, the $\alpha$ leading to best coverage ($\alpha$=5) typically has the worst performance across all alpha, why that's the case? Is the state coverage metric wrong or the assumption of better state coverage leading to better performance wrong or becase of other reasons?

Minor suggestions on writing:
1. In intro, authors motivates the usage of BE by only saying "it's widely used in behavioral economics to model human probability perception" and the results from Suresh et al. (2024). This feels void and doesn't explain why it could helps. There is no explanation why reweighting the entropy (making it more uniform or sharper) could lead to better coverage.
2. L100, "We derive practical RL methods", it's more of a different reward/objective, as you still use an existing RL algorithm.

**Questions:**

What is admissible generalized entropies and why "admissible"  and "generalized" are important?

---

> ### Author Response · Authors · 2024-11-23
> **Response to Reviewer B32C -- Part 1**
>
> We thank the reviewer for devoting their time and efforts to reading and evaluating our manuscript and we sincerely appreciate the comments and feedback provided. We are delighted that the reviewer found our work well-motivated, novel, and that they appreciated our figure design. We have thoroughly responded to all reviewer concerns below. Before addressing these in detail, we first provide general comments.
>
> **General comments:**
>
> We first note that in their weaknesses section the reviewer focused almost exclusively on the qualitative coverage experiments (t-SNE and PHATE visualizations) presented in the first half of our experimental results. We emphasize that our theoretical development and offline RL experiments make up a major part of our contribution (see general response to all reviewers). We feel that the reviewer may have undervalued these contributions.
>
> Regarding our qualitative experiments, it is important to note that **there is currently no consensus on what constitutes "good" coverage** in the unsupervised and offline RL communities. In light of this, the main point of our qualitative coverage experiments is to illustrate the *wide variety* of different kinds of coverage that can be achieved using BE compared to RE, SE, and now also RND and SMM (see Fig. 12 in the revision).
>
> In the reviewer's concerns, the reviewer appears to be thinking of coverage in distance-based, volumetric terms: that "good" coverage is achieved when points in the dataset are far apart or cover a large volume within the state space. We agree this is a potentially useful notion of coverage, so in the revision **we have proposed a volumetric coverage metric and provided a quantitative comparison of coverages provided by all datasets we considered** (see Fig. 8 in the revision). These new comparisons indicate there may be some positive correlation between volumetric coverage and downstream offline RL performance, but the results are inconclusive and merit future investigation beyond the scope of the present work.
>
>
> ---
> > **Weakness 1.1:** Could authors explain, in Fig3, why visualizing each experiment using separate plots is a valid metric of state coverage? ... As state trajectories go through non-linear projection, the distance between points more reflect their similarity rather their true distance, thus making it hard to say more spanned points covers large state space (I may be wrong on this). Regarding this, one way may be to visualize trajectories of some points and see if they are indeed very distinct states. Another way I would like to see is to run experiments on some 2d domains so we can visualize without PHATE.
>
> **Response to Weakness 1.1:**
>
> As mentioned above, the main point of our qualitative coverage experiments is to illustrate the wider variety of different coverages achievable using BE compared with RE, SE, RND, and SMM. We are not claiming that these **qualitative** results somehow provide a **quantitative** metric evaluating state space coverage. As discussed in our general remark, there is no agreed-upon quantitative metric for assessing "good" coverage.
>
> Regarding the PHATE and t-SNE plots, the reviewer is correct that the non-linear projections used in the $t$-SNE and PHATE computations are not distance preserving and the plots from Fig. 3 need not scale to reflect actual distances or volumetric notions of coverage. This is natural, however, as the main purpose of these kinds of visualization techniques is to reveal trends characteristics in data that may not be apparent in quantitative methods.
>
> Finally, to address the reviewer's concerns regarding lack of a distance- and volume-preserving quantitative coverage metric, **we have proposed a volumetric coverage metric and used it to provide quantitative coverage comparisons** in the revision (see Fig. 8). Our volumetric coverage notion considers the radius of the smallest hypersphere containing all points in the dataset (a well-studied problem in computational geometry, computed using Welzl's algorithm). See the discussions in our general remark above and following Fig. 8 in the revision for details.

---

> > ### Comment · Reviewer_B32C · 2024-11-24
> >
> > Thanks for your detailed response and extra results! Could you clarify what do you mean by "wide variety" and how Fig 3 shows that?

---

> > > ### Author Response · Authors · 2024-11-25
> > >
> > > The qualitative differences in coverage pictured in the individual plots in Fig. 3 (c.f., $\alpha = 0.2, 2.0, 5.0$) and the $t$-SNE and PHATE plots in the appendix illustrate that BE induces a wide variety of exploration behaviors for different values of $\alpha$. This is what is meant by the qualitative term "wide variety". The new volumetric coverage plots provided in Fig. 13 in the appendix of the revision also show that BE achieves a range of different volumetric coverage values, quantitatively supporting the varying levels of coverage that BE induces.
> > >
> > > We are pleased that you appreciated our detailed response and the new results we have added in response to your comments. If you have additional concerns that remain unaddressed we are happy to discuss them in the time remaining. If your concerns have been satisfactorily addressed, however, we hope you will consider increasing your score.

---

> > > > ### Comment · Reviewer_B32C · 2024-11-29
> > > >
> > > > Sorry for the late reply and thanks for the detailed response, most of my concerns are addressed and I have increased my score.

---

> ### Author Response · Authors · 2024-11-23
> **Response to Reviewer B32C -- Part 2**
>
> > **Weakness 1.2:** As each experiment is visualized separately, the PHATE may use different projections in different subfigures, making it hard to say points indeed spread out or the projection makes them look spread out. Regarding this, one more fair way is to draw PHATE plots on trajectories across all experiments to ensure they share the same projection.
>
> **Response to Weakness 1.2:**
>
> We thank the reviewer for mentioning this, as our description of the $t$-SNE and PHATE computation procedure was insufficiently clear in the original submission. When generating our visualizations we performed the $t$-SNE and PHATE plot computations for all datasets simultaneously, specifically due to the issue the reviewer has raised. **This ensures that the projections used in the visualizations are the same across all datasets, providing a fair comparison across datasets.** Since it was difficult to visualize the 19 different datasets on a single plot, we simply chose to visualize them individually as subplots, shown in Figs. 3, 9, 10, 11, 12 in the revision. We have added a footnote on page 8 of the revision clarifying this issue.
>
>
> ---
> > **Weakness 2.1:** I do not see singnificant difference between SE and BE for Quadruped coverage, is there any? If not, why that's the case?
>
> **Response to Weakness 2.1:**
>
> The reviewer raises a good point. Since PHATE and t-SNE are qualitative visualization methods involving learned, nonlinear projections, they may not reveal significant structure for all domains. In our case, we suspect that, since the dimension of the Walker environment (N=17) is much smaller than Quadruped (N=78), $t$-SNE and PHATE are simply able to discern patterns in the datasets better for Walker. In any case, the new quantitative coverage comparisons that we have provided for the rebuttal demonstrate that the diversity of quadruped datasets produced using BE is still higher than other methods in terms of variety of volumetric coverage values achieved (see Fig. 8 in the revision).
>
>
> ---
> > **Weakness 2.2:** In offline RL experiments, do you train each method until it converges? Authors mentions "we performed just 100K offline training steps", but I thought it's more fair to compare performance under convergence
>
> **Response to Weakness 2.2:**
>
> In Fig. 4 we report the maximum performance achieved by the offline RL algorithm over the course of 100K training steps. This is common practice, since overfitting the policy to the offline dataset is a persistent issue in offline RL, making simply reporting performance at the end of training not representative of actual performance achieved. To illustrate the sufficiency of 100K offline training steps for our results, **we conducted additional ablation studies** (see Fig.6 in appendix) comparing the effect of performing 100K vs. 200K offline RL training steps on performance. For these experiments, we considered 3M-element SE datasets and compared all three offline RL algorithms on all five tasks. The marginal improvements observed suggest that performing additional offline RL training has at best marginal effect on downstream task performance.
>
>
> ---
> > **Weakness 2.3:** The authors motivates in the intro that better state coverage should lead to better offline RL performance, but in Fig 6, the leading to best coverage ($\alpha=5$) typically has the worst performance across all alpha, why that's the case? Is the state coverage metric wrong or the assumption of better state coverage leading to better performance wrong or becase of other reasons?
>
> **Response to Weakness 2.3:**
>
> We again emphasize that the core message of our qualitative results is that BE provides more diverse coverage than the other methods considered. From this perspective, the difference in the coverage provided by $\alpha = 5$ compared with the coverage provided by $\alpha <= 1$ as well as that provided by SE, RE, RND, and SMM is what is important, not the downstream performance achieved for any particular parameter value. **We do not claim** in our original submission or revision that any particular parameter value or visualization will correspond to superior offline RL performance. **We do claim** that data generation methods that achieve more diverse dataset types (like the BE datasets visualized in Fig. 3) will lead to superior offline RL performance, and this hypothesis is demonstrated by the offline RL experiments presented in Fig. 4.

---

> ### Author Response · Authors · 2024-11-23
> **Response to Reviewer B32C -- Part 3**
>
> > **Suggestion 1:** In intro, authors motivates the usage of BE by only saying "it's widely used in behavioral economics to model human probability perception" and the results from Suresh et al. (2024). This feels void and doesn't explain why it could helps. There is no explanation why reweighting the entropy (making it more uniform or sharper) could lead to better coverage.
>
> **Response to Suggestion 1:**
>
> We thank the reviewer for this suggestion and have modified the introduction to provide a more detailed motivation of why BE can be expected to lead to datasets with superior coverage (see the second paragraph in the introduction, highlight in blue). We have also included a visualization and additional discussion of the BE reward function from eq. (24) in the revision (see Fig. 13) to provide further insight into how BE differs from SE and how it might be expected to affect coverage.
>
>
> ---
> > **Suggestion 2:** L100, "We derive practical RL methods", it's more of a different reward/objective, as you still use an existing RL algorithm.
>
> **Response to Suggestion 2:**
>
> We thank the reviewer for this suggestion. We have modified the second bullet point in the contributions summary in the introduction accordingly (see revision).
>
>
> ---
> > **Question 1:** What is admissible generalized entropies and why "admissible" and "generalized" are important?
>
> **Response to Question 1:**
>
> We thank the reviewer for the question and will clarify briefly here (for a detailed discussion we refer the reviewer to Suresh et al. (2024) and Amig´o et al., 2018). As discussed at the beginning of Section 2, a functional $H$ is an *admissable generalized entropy* if it satisfies the first three Shannon-Kinchin axioms, which stipulate that $H$ is (i) continuous in its parameter, (ii) uniquely maximized by the uniform distribution, and (iii) remains unchanged when known outcomes are added to the distribution. These conditions are particular important when the generalized entropy is to be used as an exploration objective, as they ensure that the functional well-behaved as an optimization objective and seeks out uniform coverage.

---

> ### Author Response · Authors · 2024-12-01
>
> Thank you very much for engaging with us during the discussion period and for raising your score. Please let us know if addressing any additional concerns could assist in improving your evaluation of our paper. We are happy to engage in further discussion if needed.

---

### Official Review · Reviewer_UCex · 2024-11-03

**Soundness:** 3
**Presentation:** 3
**Contribution:** 3
**Rating:** 6
**Confidence:** 4

**Summary:**

This paper explores how to extend Behavior Entropy to high-dimensional continuous settings and leverages this as a metric for exploration to elicit high state space coverage. This exploration objective can be systematically applied to dataset generation for offline RL algorithms. The authors additionally derive tractable KNN estimators and practical reward functions that can be used in tandem with RL to learn an effective policy. They empirically show that generated datasets with Behavior Entropy outperform datasets generated using Shannon entropy and Rényi entropy with offline RL algorithms such as CQL on Mujoco tasks.

**Strengths:**

The authors have a novel contribution with the extension of Behavior Entropy to continuous domains as well as a tractable formulation using KNNs (with convergence guarantees) to define an reward function for RL. They show initial empirical validation on a subset of the Mujoco domains in the Walker and Quadraped tasks with three offline RL algorithms: CRR, CQL and TD3. Additionally, the visualizations of the dataset generation provide insights into what differentiates the different exploration approaches.

**Weaknesses:**

Some of the empirical results aren’t compelling such as the $\alpha$/$q$ sweep done in Figure 4, where the gap from BE and Shannon/Renyi Entropy is minimal or within the margin of error for domains, which questions the advantages of the dataset generated. Additionally, it would be good to check the consistency of the dataset generation, where the dataset generation process is repeated for each $\alpha$ or $q$, $N$ times and performance is aggregated across these generation attempts. Additionally, the domains evaluated are state based and a limited subset of the Mujoco benchmark. It would be good to see the clear benefits of BE-generated datasets across multiple tasks and algorithms. Additionally, there are a myriad of other exploration objectives people consider such as RND, Count Based Exploration, etc. It would be good to empirically compare against these approaches for the synthetic dataset generation.

**Questions:**

- For BE, is there a clear selection procedure for $q$ empirically that would transfer across algorithms/tasks?
- Currently, the approach relies on a k-NN estimator, which has limitations as the state dimensionality increases or the size of the dataset is large. As offline RL scales to be used with more realistic domains, these concerns will be more common place. Are there tradeoffs with using this as an estimator?
- Along with the qualitative visualizations you have provided (e.g PHATE/tsne), could a quantitative analysis of the dataset state/action coverage be done to compare the different dataset generation approaches?

---

> ### Author Response · Authors · 2024-11-23
> **Response to Reviewer UCex -- Part 1**
>
> We thank the reviewer for devoting their time and effort to evaluating our submission, and we sincerely appreciate the insightful feedback provided. We are pleased that the reviewer recognized the novelty of our work and appreciated the insights provided by our experimental results. We have thoroughly responded to each of the reviewer's concerns below.
>
> **Insufficient experiments issue:**
>
> Before addressing each of the reviewer's concerns in detail, we first highlight that one of the reviewer's primary concerns is that insufficient experiments were conducted (three out of four weaknesses described by the reviewer relate to this). The reviewer proposes several ways to expand the experiments to remedy this, including: (i) replication of dataset generation, (ii) additional data generation algorithms (RND, etc.), (iii) additional environments, (iv) additional offline RL algorithms. While we agree that expanding the experiments to include all these desiderata is ideal and would make the experimental results  extremely robust, we emphasize that the computational expense this would incur is extreme. We note that **we have performed additional experiments and significantly revised our submission to address issue (ii)**, as detailed in the general response to all reviewers, but we argue below that **requiring (i), (iii), and (iv) as well is computationally infeasible.**
>
> To provide the reviewer with additional insight into the computational effort involved, we present details in table form below. First, consider the following table adding up the total number of individual trials required to perform the offline RL training portion of the experiments (illustrated in Fig. 6 in the appendix) in our original submission.
>
>
> | Data generation methods | Envs | Downstream tasks (per envrionment) | Offline RL algorithms | Offline RL seeds | Total trials |
> |----------|----------|----------|----------|----------|----------|
> | 14     | $\times 1$    | $\times 2$ (Quadruped)   | $\times 3$ | $\times 5$ | $= 420$ |
> | 14     | $\times 1$    | $\times 3$ (Walker)   | $\times 3$ | $\times 5$ | $= 630$ |
>
> Each of the 1050 (= 420 + 630) trials required a training time of approximately 2 GPU hours on our hardware. For the dataset generation portion of our experiments, we trained 28 (= data generation methods $\times$ environments) separate data generation policies, each of which required 8 GPU hours. In total, approximately 2324 GPU training hours were required for our experiments.
>
> To achieve desiderata (i)-(iv) above, suppose that we: (i) repeat data generation $N = 5$ times for each data generation method; (ii) include RND [Burda et al., 2019], SMM [Lee et al., 2019], and DIAYN [Eysenbach et al., 2019]; (iii) consider 2 additional environments with 3 downstream tasks apiece; (iv) evaluate 2 additional offline RL algorithms. The resulting number of offline RL trials is given in the following table.
>
> | Data generation methods | Data generation replications | Envs | Tasks (per env) | Offline RL algorithms | Offline RL seeds | Total trials |
> |----------|----------|----------|----------|----------|----------|----------|
> | 17     | $\times 5$    | $\times 1$    | $\times 2$  | $\times 5$ | $\times 5$ | $= 4250$ |
> | 17     | $\times 5$    | $\times 3$    | $\times 3$  | $\times 5$ | $\times 5$ | $= 19125$ |
>
> This gives a total of 23375 (= 4250 + 19125) distinct trials. Including the 340 (= data generation methods $\times$ data generation replications $\times$ total number of environments) data generation policies that need to be trained and assuming compute times are similar to the experiments we ran for the original submission, **this would entail 49470 GPU hours of training** altogether. We believe that it is unrealistic to expect this level of experimental cost even from a primarily experimentally focused submission. Given that **core parts of our overall contribution are theoretical and algorithmic** (see Sections 2-4), we maintain that this is especially true in our case.

---

> ### Author Response · Authors · 2024-11-23
> **Response to Reviewer UCex -- Part 2**
>
> > **Weakness 1:** Some of the empirical results aren’t compelling such as the $\alpha/q$ sweep done in Figure 4, where the gap from BE and Shannon/Renyi Entropy is minimal or within the margin of error for domains, which questions the advantages of the dataset generated.
>
> **Response to Weakness 1:**
>
> The core message of our offline RL results is that BE datasets provide at best superior and at worst competitive offline RL performance on downstream tasks when compared with alternative dataset generation methods, and that BE is therefore an important new exploration objective to consider when performing data generation for offline RL. We respectfully disagree with the reviewer that the experiments presented in Figures 4 (and 5) are not compelling, **especially in light of the additional experiments performed for the revision.** As summarized in Table 1 and discussed in Section 5, BE datasets lead to superior performance over SE, RND, and SMM datasets across all tasks, while BE datasets lead to superior performance over RE datasets on 4 out of 5 tasks and remain competitive on the fifth task. As summarized on lines 511-514 of our submission, in our experiments
>
> > "best offline RL performance on BE-generated datasets clearly exceeds best performance on RE datasets on 13 out of 15 task-algorithm combinations and best performance on SE, RND, and SMM datasets on 15 out of 15 task-algorithm combinations."
>
> Given that offline RL training for each task-algorithm combination was evaluated over 5 independent replications, we emphasize that **the superior offline RL performance observed on BE-generated datasets is statistically meaningful** and that the usefulness of BE as an exploration objective for dataset generation is empirically well-supported by our experiments. In addition, in our revision we have shown that this trend continues to hold for additional dataset generation methods (see Figures 4 and 5 in the revision).
>
>
> ---
> > **Weakness 2:** Additionally, it would be good to check the consistency of the dataset generation, where the dataset generation process is repeated for each $\alpha$ or $q$, $N$ times and performance is aggregated across these generation attempts.
>
> **Response to Weakness 2:**
>
> While we agree with the reviewer that verifying the consistency of the data generation process in this way would be ideal, we emphasize that the computational burden that this entails for statistically meaningful values of $N$ (e.g., $N \geq 5)$ is prohibitive. This limitation is especially acute in our case, as the number of data generation methods we considered in the original submission is 14 (each $\alpha$ and $q$ value corresponds to a distinct dataset generation method) which is large compared to previous work: only 8 dataset generation methods were considered in both the URLB [Laskin et al., 2021] and ExORL [Yarats et al., 2021] benchmarks. The reasons for this are outlined in the **Insufficient experiments issue** discussion above. We furthermore highlight that the $N=1$ approach to dataset generation that we have followed is standard in the unsupervised RL literature due to precisely the same practical limitations that we have described (see, e.g., the dataset generation procedures used in URLB and ExORL).
>
>
> ---
> > **Weakness 3:** Additionally, the domains evaluated are state based and a limited subset of the Mujoco benchmark. It would be good to see the clear benefits of BE-generated datasets across multiple tasks and algorithms.
>
> **Response to Weakness 3:**
>
> We again agree with the reviewer that considering additional environments would be ideal. Nonetheless, we reiterate that the computational burden this entails is prohibitive for the present work. Specifically, notice that for each additional environment that we consider we need to generate new datasets for each data generation method, then train multiple independent replications of each offline RL algorithms on each downstream task on that environment. Assuming 14 dataset generation methods, 3 downstream tasks, 3 offline RL algorithms, and 5 offline RL seeds, this entails $675 = 15 \times 3 \times 3 \times 5$ additional trials for each new environment considered.

---

> ### Author Response · Authors · 2024-11-23
> **Response to Reviewer UCex -- Part 3**
>
> > **Weakness 4:** Additionally, there are a myriad of other exploration objectives people consider such as RND, Count Based Exploration, etc. It would be good to empirically compare against these approaches for the synthetic dataset generation.
>
> **Response to Weakness 4:**
>
> Thanks to the reviewer for raising this valid point. Importantly, it is also computationally feasible, since adding a single additional data generation method to our original experiment setup (c.f., Figure 4) only results in an additional 75 trials. **We have therefore performed additional offline RL experiments on both RND- and SMM-generated datasets as well as a 3M-element SE-generated dataset** for the rebuttal and have provided an updated revision presenting these results (see Figures 4 and 5 in the revision). We hope that these additional results satisfy the reviewer, but we also emphasize that our original submission did compare our BE-maximizing approach with the popular SE-maximizing approach of APT (the specific algorithm we compare against is the ICM-based implementation of APT, as described in Section 5) as well as an RE-maximizing variant of this algorithm. We believe that the revised experiments strengthen the contribution of our paper and again thank the reviewer for suggesting this improvement.
>
>
> ---
> > **Question 1:** For BE, is there a clear selection procedure for $q$ empirically that would transfer across algorithms/tasks?
>
> **Response to Question 1:**
>
> The question of how to select the best value of $\alpha$ (for BE) or $q$ (for RE) for optimal data generation is a good one. For our experiments we have simply considered an evenly distributed range of $\alpha$ and $q$ values and we have observed empirically that values $\alpha, q < 1$ lead to superior offline RL performance on downstream tasks. Without prior information about the domain and the specific downstream tasks to be solved, it is currently unclear how $\alpha$ and $q$ might be selected more systematically. However, we believe that with either prior information about the domain and downstream tasks or feedback during the unsupervised RL portion of training, it should be possible to adapt the values of $\alpha$ and $q$ during data generation to provide superior coverage for learning the downstream tasks at hand. This is an interesting direction for future work.
>
>
> ---
> > **Question 2:** Currently, the approach relies on a k-NN estimator, which has limitations as the state dimensionality increases or the size of the dataset is large. As offline RL scales to be used with more realistic domains, these concerns will be more common place. Are there tradeoffs with using this as an estimator?
>
> **Response to Question 2:**
>
> There are two main reasons why the use of $k$-NN estimation remains computationally tractable even for large datasets and high-dimensional state spaces. **Concerning large datasets,** the $k$-NN computation is carried out on each mini-batch sampled from the replay buffer. This is in keeping with the APT [Liu & Abbeel, 2021] and Proto-RL [Yarats et al., 2021] algorithms, and, as discussed in Section 5 starting around line 386 of our original submission, we used APT to maximize BE (as well as RE and SE). Since mini-batch sizes were fixed at $1024$ (see Table 2 in the appendix), the computational cost of computing $k$-NNs for $k = 12$ (this $k$ was chosen to remain consistent with the ExORL framework) is fairly small. **Regarding high-dimensional state spaces,** one of the key features of the APT method is that it learns a feature mapping into a feature space of *user-defined dimension*, then performs $k$-NN estimation over the resulting feature space. This allows the APT method to handle potentially high-dimensional state spaces via user selection of an appropriate feature space dimension and, since we use APT as our base algorithm, we enjoy this feature as well.

---

> ### Author Response · Authors · 2024-11-23
> **Response to Reviewer UCex -- Part 4**
>
> > **Question 3:** Along with the qualitative visualizations you have provided (e.g PHATE/tsne), could a quantitative analysis of the dataset state/action coverage be done to compare the different dataset generation approaches?
>
> **Response to Question 3:**
>
> The reviewer raises an important open problem in the unsupervised RL and offline RL literatures: **there is currently no consensus in these communities on the best way to quantitatively evaluate how "good" an exploration objective is or how "good" the coverage of a given dataset is.** One symptom of this is the proliferation of different exploration objectives in unsupervised RL: if there were agreement on the best metrics for quantifying the goodness of exploration (e.g., Shannon entropy), then exploration methods would naturally focus on optimizing those metrics; this is not the case, however, highlighting the aforementioned lack of consensus in the unsupervised RL community. In the offline RL setting, many offline RL datasets exist, but the methods used to generate the datasets vary widely. In the widely used D4RL benchmark [Fu et al., 2021, https://arxiv.org/pdf/2004.07219], for example, due to lack of an accepted metric for evaluating dataset coverage, a hodgepodge of random, partially trained, and expert policies are used for dataset generation (see the appendix of [Fu et al., 2021] for a description of the domains and dataset generation methods used). This lack of principled methods for dataset generation for offline RL is one of the primary motivations underlying the development of the ExORL framework.
>
> Despite this question remaining an open problem, to more fully address the reviewer's concerns in our rebuttal **we have proposed a quantitative metric for evaluating dataset coverage and used this metric to provide a quantitative comparison of the datasets we considered.** The quantitative metric we propose is the volume of the smallest hypersphere covering the dataset, which we compute using Welzl's algorithm, a standard procedure from the computational geometry literature for solving the smallest-circle problem. We have included these quantitative comparison and a discussion of them in Figure 8 in the appendix of our revision.

---

> ### Author Response · Authors · 2024-11-25
>
> Thanks again for your time and effort in reviewing our paper. We are following up to ensure that our detailed response and revision have addressed all your concerns. We are happy to address any remaining questions before the end of the discussion period tomorrow.

---

> > ### Comment · Reviewer_UCex · 2024-11-27
> > **Response to Authors**
> >
> > Thank you for the clarifications you have provided. I understand and appreciate that conducting empirical experiments is time-consuming and may not be practical within the rebuttal period, given the additional experiments already included. While I remain concerned about the performance of this approach in more complex tasks and high-dimensional state settings, I acknowledge and value the theoretical contributions of this work. Accordingly, I have decided to raise my score to a 6.

---

> ### Author Response · Authors · 2024-12-01
>
> Thank you very much for engaging with us during the discussion period and for raising your score. Please let us know if addressing any additional concerns could assist in improving your evaluation of our paper. We are happy to engage in further discussion if needed.

---

### Official Review · Reviewer_esfg · 2024-11-04

**Soundness:** 2
**Presentation:** 3
**Contribution:** 2
**Rating:** 6
**Confidence:** 3

**Summary:**

This paper studies dataset generation for offline RL. More specifically, this paper extends recently proposed behavioral entropy (BE) for discrete settings to continuous tasks, and designs estimator and RL algorithms for exploration and data generation. The paper compares offline RL performance on BE-generated data with other entropy-based data generation and shows that the proposed method outperforms other baselines in the Walker and Quadruped environments.

**Strengths:**

1. The proposed method is well-motivated and theoretically grounded.
2. The writing is clear and easy to follow.

**Weaknesses:**

While well-motivated, I have some questions about this work:

1. The authors propose a KNN-based estimator with convergence guarantee. I wonder in practice how $k$ is chosen and if KNN introduces huge computation overhead compared to other baselines. For instance, if SE and RE have the same computation budget as BE, will they collect more data and hence perform better?
2. The experiments only generate 500K elements for each task, which might be not enough for the offline RL to learn a good policy. Can we test if the performance of BE, SE, and RE will scale with more data?

Minor:

1. While this paper focuses on the offline RL setting, I am curious about whether this entropy-based exploration can be applied in the online RL setting and how it performs compared with other online exploration methods.

Overall, while this paper presents some interesting ideas, I am unable to recommend acceptance at this stage given the questions mentioned above. However, I would consider raising my score if the authors could address my concerns.

**Questions:**

There are some questions and concerns, which I have outlined in the previous section.

---

> ### Author Response · Authors · 2024-11-23
> **Response to Reviewer esfg**
>
> We thank the reviewer for devoting their time and effort to evaluating our submission, and we sincerely appreciate the insightful feedback provided. We are pleased that the reviewer found our contribution well-motivated, theoretically grounded, and well-presented. We have thoroughly responded to each of the reviewer's concerns below.
>
>
> ---
> > **Weakness 1:** The authors propose a KNN-based estimator with convergence guarantee. I wonder in practice how $k$ is chosen and if KNN introduces huge computation overhead compared to other baselines. For instance, if SE and RE have the same computation budget as BE, will they collect more data and hence perform better?
>
> **Response to Weakness 1:**
>
> We thank the reviewer for raising these important questions. **Regarding the choice of $k$,** in our experiments we chose $k = 12$ (see Table 2 in the appendix) as used in the ExORL framework [Yarats et al., 2022] to ensure that our experiments remained comparable with benchmark APT implementation considered in that framework. In general, however, the choice of $k$ is a matter of trial and error: early work on $k$-NN estimators for Shannon entropy experimentally found $k = 4$ to lead to reasonable performance [Singh et al., 2003], while the paper proposing APT [Liu and Abbeel, 2021] used values $k = 3, 5,$ and $10$ depending on the environment. From what we know of the literature on $k$-NN entropy estimators, relatively small values $k \leq 15$ are most common, even in high-dimensional spaces.
>
> **Concerning the computational overhead from using $k$-NN estimation,** it is important to note that the $k$-NN computation is carried out on each mini-batch sampled from the replay buffer. This is in keeping with the APT and Proto-RL [Yarats et al., 2021] algorithms, and, as discussed in Section 5 starting around line 386 of our submission, we used APT to maximize behavioral entropy (BE), Renyi entropy (RE), and Shannon entropy (SE). Since mini-batch sizes were fixed at $1024$ (see Table 2 in the appendix), the **computational cost of computing $k$-NNs for $k = 12$ is small.** We emphasize that SE and RE were estimated using the same underlying $k$-NN computation as that used in BE, ensuring a fair comparison between all three methods. In particular, this means that all three methods compared (SE, RE, BE) had the same computation budget.
>
>
> ---
> > **Weakness 2:** The experiments only generate 500K elements for each task, which might be not enough for the offline RL to learn a good policy. Can we test if the performance of BE, SE, and RE will scale with more data?
>
> **Response to Weakness 2:**
>
> The reviewer is right that we only generated 500K-element datasets for each combination of exploration algorithm and environment, which contrasts sharply with the large (10M+ element) datasets often considered in offline RL settings. However, as described in Section 5, lines 385-391, despite using only 500K-element datasets we achieved comparable performance on downstream tasks to benchmark algorithms trained on 10M-element datasets in the ExORL framework. Importantly, the superior downstream performance on BE datasets compared with SE and RE datasets that we observed indicates that BE can be used to achieve better data- and sample-efficiency when used to perform dataset generation for offline RL (see lines 389-391). This fact illustrates a key strength of our approach. To highlight this feature of our approach while simultaneously addressing the concerns of the reviewer, for the rebuttal **we have performed additional offline RL experiments on a 3M-element SE-generated dataset** and have revised the submission accordingly (see Figs. 4, 5). These experiments demonstrate that 500K-element datasets generated using BE continue to lead to superior downstream offline RL performance than 3M-element datasets generated using SE.
>
>
> ---
> > **Question 1:** While this paper focuses on the offline RL setting, I am curious about whether this entropy-based exploration can be applied in the online RL setting and how it performs compared with other online exploration methods.
>
> **Response to Question 1:**
>
> We agree with the reviewer that BE-based exploration for the online RL setting is an interesting direction worth further investigation. Given our focus on data generation for offline RL, however, we believe this important question is out of scope for the present submission and best left for future work.

---

> ### Author Response · Authors · 2024-11-25
>
> Thanks again for your time and effort in reviewing our paper. We are following up to ensure that our response and revision have addressed all your concerns. We are happy to address any remaining questions before the end of the discussion period tomorrow.

---

> > ### Comment · Reviewer_esfg · 2024-11-26
> > **Thanks for the response**
> >
> > I appreciate the reviewer for providing clarification and additional experimental results to address my concerns. I have two follow-up questions regarding Weakness 2:
> > 1. Given that the 3M-SE experiment has both positive and negative results depending on the task, does this issue also exist for the proposed BE method?
> > 2. I also noticed that the ratio between the dataset and training steps in the experiment is much different from that of ExORL, does this significantly affect the final performance of all baselines?

---

> > > ### Author Response · Authors · 2024-11-30
> > >
> > > We are pleased that the reviewer appreciated our response and additional experiments. Regarding the follow-up questions:
> > > 1. We believe the reviewer is referring to the fact that, in Fig. 6, on some downstream tasks performing 200K offline training steps actually results in worse performance than using only 100K offline training steps (please let us know if we have misunderstood the question). This is a good observation, and we also noticed this fact. It is important to note that, given how close the average values were in every case, we suspect that this is simply due to only five independent trials being used for this experiment; if a very large number of trials were performed, we expect that 200K would tend to always outperform 100K. **Regarding whether this holds in the BE case:** we did not perform this ablation for the BE datasets due to the large number of different ablations that would need to be performed (one for each $\alpha$ value) for the experiment to be meaningful. For insight into the computational expense that was required for our experiments, please see the tables in our response to reviewer UCex.
> > > 2. The surprising thing is that, despite the dataset sizes and number of offline RL training steps being much smaller in our case than in the ExORL setup, downstream performance was largely comparable to that achieved in the ExORL setting. This is discussed in lines 387-392 in the revision. This highlights the importance of developing methods like BE to generate high quality datasets for offline RL, as they can lead to improved data- and sample-efficiency.
> > >
> > > Thanks again for your time and effort, and we hope that our responses and revision have been able to address all your concerns.

---

> ### Author Response · Authors · 2024-12-01
>
> Thank you very much for engaging with us during the discussion period. Please let us know if addressing any additional concerns could assist in improving your evaluation of our paper. We are happy to engage in further discussion before the end of the discussion period tomorrow, if needed.

---

> > ### Comment · Reviewer_esfg · 2024-12-01
> > **Thanks for the clarification**
> >
> > Thank you for the clarification. I have raised my score accordingly.

---

> > > ### Author Response · Authors · 2024-12-02
> > >
> > > Thank you very much for engaging with us during the discussion period and for raising your score. Please let us know if addressing any additional concerns could assist in improving your evaluation of our paper. We are happy to engage in further discussion if needed.

---

### Author Response · Authors · 2024-11-23
**General response to all reviewers**

We thank all the reviewers for their valuable feedback and insightful questions. We are encouraged that they consider our work well-motivated and novel (esfg, UCex, B32C), and appreciated our theoretical contributions (esfg, UCex) and the clarity of the writing and figures (esfg, UCex, B32C, T8Fn).

**Revision and new experiments:**

We have uploaded a revised version of our paper based on reviewer suggestions. Our main changes are highlighted in blue. The primary additions are as follows:
* new **offline RL experiments on datasets generated using RND, SMM,** and a 3M-element SE dataset, including $t$-SNE and PHATE visualizations (esfg, UCex, T8Fn) -- see Table 1 and Figs. 4, 5, 6, 12
* new **quantitative comparisons of dataset coverage** (UCex, B32C, T8Fn) -- see Fig. 8
* **clarification of the motivation for using BE for exploration** (B32C, T8Fn) -- see intro, Figs. 8, 13

**Summary of core contributions:**

1. we extend the recently proposed BE to continuous spaces **(theoretical)**
2. we derive $k$-NN estimators for BE, rigorously establish convergence guarantees for these estimators, and derive practical RL rewards based on these $k$-NN estimators for use in maximizing BE **(theoretical)**
3. we provide qualitative and quantitative comparisons of coverage provided by BE and other methods, illustrate superior variety of coverage provided by BE **(experimental)**
4. we provide quantitative comparison of offline RL performance on datasets generated using BE, RE, SE, RND, and SMM, and find that BE datasets generally lead to superior (or comparable in the worst case) performance on downstream tasks **(experimental)**

**Final remarks:**

The reviewers focused heavily on our experimental results in their reviews, yet key parts of our contribution are theoretical (see contributions above). **We believe the reviewers may have undervalued our theoretical contributions,** and we hope the reviewers will take this into consideration during the discussion period and revise their scores where appropriate.

We address specific reviewer concerns in individual responses below. We hope that our revisions, additional experiments, and individual responses have addressed all reviewer questions and concerns. Please let us know if there are any further questions.

---

### Meta-Review · Area_Chair_mjN5 · 2024-12-20

**Metareview:**

This paper introduces a novel extension of behavioral entropy (BE) to continuous state spaces for exploration and dataset generation in offline reinforcement learning (RL). By proposing a KNN-based estimator, the authors show that BE-generated datasets outperform those based on Shannon and Rényi entropy, leading to improved offline RL performance on Mujoco tasks.

While the work is generally seen as a valuable theoretical contribution, there are concerns about its practical applicability. Introducing environment-specific hyperparameters (e.g., α and q) requires tuning for each task, which could be cumbersome in offline RL settings. The computational complexity of the KNN estimator, especially in high-dimensional spaces, was also noted. Despite these concerns, the authors clarified that the method works well in practice, though further explanation of computational costs could be useful. Some reviewers also pointed out that BE improved state coverage but did not consistently lead to better performance compared to simpler methods like Shannon entropy or strong offline methods like TD3.

Overall, the rebuttal addressed many of the reviewers' concerns, improving the clarity of the paper. Given the theoretical contributions and promising results, I recommend acceptance. However, the authors must clarify practical aspects such as hyperparameter tuning and computational cost in the final version.

**Additional Comments On Reviewer Discussion:**

During the rebuttal period, reviewers raised concerns about the hyperparameter sensitivity, the computational complexity of the KNN estimator, and the lack of comparison to simpler exploration methods like RND. They also questioned the relationship between state coverage and offline RL performance.

The authors addressed these concerns by clarifying the process of selecting hyperparameters, explaining that while tuning is required, it is a common practice in RL. They also acknowledged the computational cost of the KNN estimator but explained that it works well in practice despite not detailing the trade-offs. Additional experimental results comparing BE to other entropy-based methods were provided, and the authors clarified the relationship between state coverage and RL performance, noting that factors beyond state coverage influence offline RL results.

While there are still concerns about hyperparameter tuning and computational complexity, the authors have significantly improved the paper. The contributions are valuable, and the empirical results are promising.

---

### Decision · Program_Chairs · 2025-01-22

Accept (Poster)